# VN-EGNN: E(3)- and SE(3)-Equivariant Graph Neural Networks with Virtual Nodes Enhance Protein Binding Site Identification

## Abstract

Being able to identify regions within or around proteins, to which ligands can potentially bind, is an essential step in developing new drugs. Binding site identification methods can now profit from the availability of large amounts of 3D structures in protein structure databases or from AlphaFold predictions. Current binding site identification methods heavily rely on graph neural networks (GNNs), usually designed to output E(3)-equivariant predictions. Such methods turned out to be very beneficial for physics-related tasks like binding energy or motion trajectory prediction. However, the performance of GNNs at binding site identification is still limited potentially due to a lack of expressiveness capable of modeling higher-order geometric entities, such as binding pockets. In this work, we extend E($n$)-equivariant graph neural networks (EGNNs) by adding virtual nodes and applying an extended message passing scheme. The virtual nodes in these graphs are dedicated entities to learn representations of binding sites, which leads to improved predictive performance. In our experiments, we show that our proposed method, VN-EGNN, sets a new state-of-the-art at locating binding site centers on COACH420, HOLO4K and PDBbind2020.

## 1 Introduction

**Binding site identification remains a central computational problem in drug discovery.** With the advent of AlphaFold (Jumper et al., 2021; Abramson et al., 2024), millions of 3D structures of proteins have been unlocked for further investigation by the scientific community (Tunyasuvunakool et al., 2021; Cheng et al., 2023). The 3D structure of a protein can provide crucial information about its function, and drug discovery is one of the most important fields that profits from these 3D structures (Ren et al., 2023; Sadybekov and Katritch, 2023). It has been envisioned that the availability of 3D structures will allow to purposefully design drugs that alter protein function in a desired way. However, to enable structure-based drug design, further computational approaches, such as *docking* or *binding site identification* methods, have to be employed (Lengauer and Rarey, 1996; Cheng et al., 2007; Halgren, 2009). While docking approaches predict the location of a specific small molecule, called a ligand, within a protein's active site upon binding, binding site identification aims at finding regions on the protein likely to form a binding pocket and interact with unknown ligands (Schmidtke and Barril, 2010). Note that docking and binding site identification are fundamentally different tasks in structure-based drug design: for the vast majority of proteins no ligand is known, and binding site identification methods can provide valuable information for understanding protein function, guiding rational drug design or identifying a protein as a potential drug target. For both approaches, deep learning methods, and specifically geometric deep learning have brought significant advances (Gainza et al., 2020; Sverrisson et al., 2021; Méndez-Lucio et al., 2021; Ganea et al., 2022; Stärk et al., 2022; Lu et al., 2022; Corso et al., 2023).

**Methods for binding site identification.** The identification of binding sites relies on the successful combination of physical, chemical and geometric information. Initially, machine learning methods for binding site prediction were based on carefully designed input features due to their tabular processing structure. For instance, FPocket (Le Guilloux et al., 2009) relies on Voronoi tessellation and alpha spheres (Liang et al., 1998) and additionally takes an electronegativity criterion into account. P2Rank, a random-forest-based method, makes use of the protein surface (Krivák and Hoksza, 2018). With

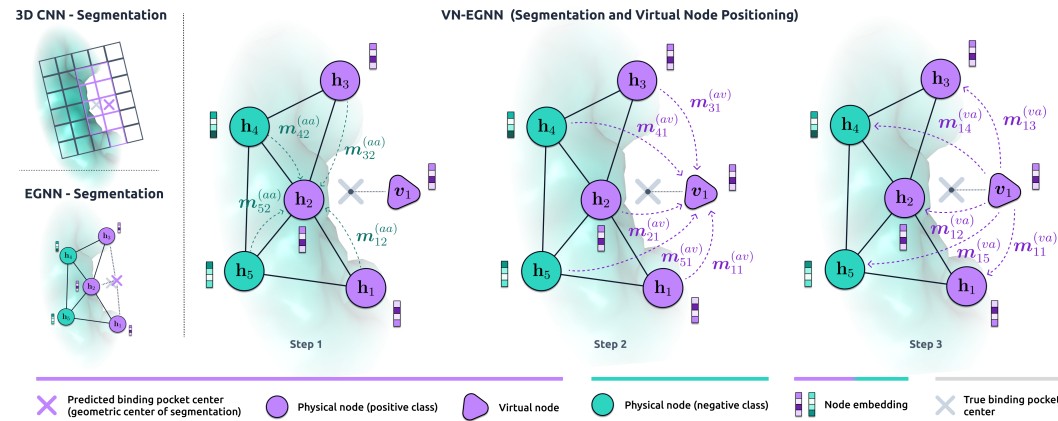

Figure 1: Overview of binding site identification methods. **Top Left**: Traditional methods, based on segmentation of a voxel grid, in which the pocket center is calculated as the geometric center of the positively labeled voxels. **Bottom Left:** Geometric Deep Learning approaches, such as EGNN, in which the pocket center is calculated as the geometric center of the positively labeled nodes. **Right:** VN-EGNN approach (ours): the predicted binding site center is the position of the virtual node after $L$ message passing layers.

the advent of end-to-end deep learning and especially with the breakthrough of convolutional (Lecun et al., 1998) and graph neural networks (GNNs) (Scarselli et al., 2009; Defferrard et al., 2016; Kipf and Welling, 2017; Gilmer et al., 2017; Satorras et al., 2021), the construction of input features can be learned which helped to advance predictive quality. For instance, DeepSite (Jiménez et al., 2017) is a voxel-based 3D convolutional neural network for binding site prediction. Convolutional operations on the 3D space are, however, computationally very demanding and so quickly other approaches to tackle binding site identification were developed, e.g., DeepSurf (Mylonas et al., 2021) or PointSite (Yan et al., 2022). DeepSurf operates on surface-based representations and places several voxelized grids on the protein's surface, while PointSite is based on a form of sparse convolutions to reduce the computational overhead and keep sparse regions in the 3D space sparse. Typical convolutional networks, however, do not perform well at binding site identification, likely because of the irregularity of protein structures and due to the fact that proteins may be arbitrarily rotated and shifted in space (Zhang et al., 2023b). Thus, geometric deep learning approaches, most notably (graph-based) group-equivariant architectures, such as EquiPocket (Zhang et al., 2023b), which are equivariant to the group of Euclidean transformations in 3D space (E(3)), are powerful methods for binding site identification.

**E(3)-equivariant graph neural networks.** We use graph neural networks (GNNs) that are robust to transformations of the Euclidean group, i.e., rotations, reflections, and translations, as well as to permutations. From a technical point of view, equivariance of a function $f$ to certain transformations means that for any transformation parameter $g$ and all inputs $x$ we have $f(T_g(x)) = S_g(f(x))$, where $T_g$ and $S_g$ denote transformations on the input and output domain of $f$, respectively (see App. D for further information). Equivariant operators applied to molecular graphs allow to preserve the geometric structure of the molecule. We build on E($n$)-equivariant GNNs (EGNNs) of Satorras et al. (2021) applied to three dimensional space and the problem of binding site identification. In contrast to methods such as MACE (Batatia et al., 2022), Nequip (Batzner et al., 2022), or SEGNN (Brandstetter et al., 2021), EGNNs operate on scalar features, e.g., distances, and use scale operations for coordinate updates. Thus, EGNNs operate efficiently (Villar et al., 2021) without resorting to compute-expensive higher order features, and, most importantly, allow for efficient coordinate update of virtual nodes.

**Limitations of GNNs and a mitigation strategy.** Graph neural networks can suffer from limited expressiveness (Morris et al., 2019; Xu et al., 2019), oversmoothing (Li et al., 2018; Rusch et al., 2023), or oversquashing (Alon and Yahav, 2021; Topping et al., 2022), which can lead to unfavorable learning dynamics or weak predictive performance. To improve the learning dynamics of GNNs, several works have introduced virtual nodes, sometimes called super-nodes or supersource-nodes,

that are introduced into a message passing scheme and connected to all other nodes. In a benchmark setting, Hu et al. (2020) and Rosenbluth et al. (2024) showed that adding virtual nodes tends to increase the predictive performance. Hwang et al. (2022) provide a theoretical analysis of the benefits of virtual nodes in terms of expressiveness, demonstrate the increased expressiveness of GNNs with virtual nodes and also hint at the fact that such nodes can decrease oversmoothing. Alon and Yahav (2021) mention that virtual nodes might be used as a technique to overcome oversquashing effects. Cai et al. (2023) and Cai (2023) show that an MPNN with one virtual node, connected to all nodes, can approximate a Transformer layer. Low rank global attention (Puny et al., 2020) can be seen as one virtual node, which improves expressiveness. Practically, virtual nodes have already been suggested in the original work by Gilmer et al. (2017), and they were even mentioned earlier in Scarselli et al. (2009) and used in application areas such as drug discovery (Li et al., 2017; Pham et al., 2017; Ishiguro et al., 2019). Joshi et al. (2023) investigated the expressive power of EGNNs in greater detail and argue that these networks can suffer from oversquashing. In order to alleviate the oversquashing problem of EGNNs for binding site identification, we suggest to extend EGNNs with virtual nodes and introduce an adapted message passing scheme. We refer to this method as Virtual-Node Equivariant GNN (VN-EGNN). A related method (MEAN) to ours, which is in the context of EGNNs and which uses global nodes, that are connected to many other graph nodes (i.e., all within components), was suggested by Kong et al. (2023) for conditional antibody design. MEAN (applied to components) can potentially be considered to be the first EGNN-like architecture using virtual nodes.

**EGNNs with virtual nodes for binding site identification.** In accordance with previous approaches, we consider binding site identification as a segmentation task. While other methods are, e.g., based on voxel grids, our method is based on EGNNs with virtual nodes, where all atoms or residues of the protein (the physical entities) are represented by physical, i.e., non-virtual, nodes in the graph (see Fig. 1, left). The objective is to correctly classify whether a node is within a certain radius of a region to which potential ligands can bind. Therefore, binding site identification can be considered as a node-level binary classification task and thus a semantic segmentation task. For this task, the ground truth is whether an atom was within a certain radius to experimentally observed protein binding ligands. In addition to node features, EGNNs act on coordinate features associated with each node, and both feature types are updated during message passing. While it appears straightforward to associate physical nodes with the protein's atoms, it is a-priori unclear if the coordinate embeddings of virtual nodes are useful for the task at hand. In an initial experiment, we trained VN-EGNN to learn a semantic segmentation task using multiple virtual nodes to which we assigned random coordinates. In an analysis of the results, we could empirically observe that coordinates of the virtual nodes converged towards the actual physical binding positions of ligands on the protein (see App. H.1 for further details and Fig. 1, right, for a visualization). The results of this initial experiment gave rise to the assumption that virtual nodes enable VN-EGNNs to form useful neural representations of binding sites and especially allow the prediction of locations of binding site centers. This, however, further implies that the binding site center itself may be a useful optimization target to train VN-EGNNs. Thus, we extended our objective from only predicting whether physical nodes are close to binding regions, to also directly taking the distance between observed and predicted binding site centers into account. With this multi-modal objective, the coordinate embeddings of virtual nodes are trained to predict the locations of binding site centers. The remaining features of the virtual nodes are considered to form an abstract neural representation of a protein binding site.

**Contributions.** In this work, we aim at improving binding site identification through geometric deep learning methods. Here, we follow the approach of using EGNNs (Satorras et al., 2021; Zhang et al., 2023b) for identifying binding pockets. Although EGNNs are prime candidates for this task, traditional EGNNs exhibit poor performance at binding site identification (Zhang et al., 2023b), which might be due to a) their lack of dedicated nodes that can learn representations of binding sites, and b) oversquashing effects which hamper learning (Alon and Yahav, 2021; Topping et al., 2022; Joshi et al., 2023). We aim to alleviate both problems by using EGNNs with virtual nodes. In this work, we contribute the following:

- We propose to adapt E(3)-equivariant GNNs towards the identification of binding sites of proteins.

- We demonstrate that the virtual nodes in the message passing scheme learn useful representations and accurate locations of binding pockets.

- We assess the performance of other methods, baselines and our method on benchmarking datasets.

## 2 E(3)-Equivariant Graph Neural Networks with Virtual Nodes

### 2.1 Notational Preliminaries

We give an overview on variable and symbol notation in App. B, and a more detailed description and discussion on how we represent proteins and binding sites in Apps. C.1 and C.2, respectively. To quickly summarize, the coordinates of the $i$-th *physical node*, e.g., the location of an atom of a protein, are denoted as $\mathbf{x}_i \in \mathbb{R}^3$, and its other node features as $\boldsymbol{h}_i \in \mathbb{R}^D$. $l$ added as an index to symbols will indicate neural network layers, but might be omitted for simplicity sometimes if it is clear from the context. We will consider *virtual nodes* and the $k$-th virtual node coordinates will be denoted as $\mathbf{z}_k$, while the other virtual node features will be denoted as $\boldsymbol{v}_k$. We use upper-case bold letters to denote the matrices collecting the coordinates and features of the $N$ physical and the $K$ virtual nodes, respectively: $\boldsymbol{H}^l := (\boldsymbol{h}_1^l, \ldots, \boldsymbol{h}_N^l)$, $\mathbf{X}^l := (\mathbf{x}_1^l, \ldots, \mathbf{x}_N^l)$, $\boldsymbol{V}^l := (\boldsymbol{v}_1^l, \ldots, \boldsymbol{v}_K^l)$, $\mathbf{Z}^l := (\mathbf{z}_1^l, \ldots, \mathbf{z}_K^l)$.

We denote the graph, which VN-EGNN works upon as $\mathcal{G}$ and $\boldsymbol{A}$ as the associated adjacency matrix. $\mathcal{N}(i)$ indicates neighbouring nodes to node $i$ within $\mathcal{G}$ and edge features between nodes $i$ and $j$ as $a_{ij}$ (for two non-virtual nodes) and $d_{ij}$ (between physical and virtual nodes). At training time, we have access to node-level labels $y_i \in \{0, 1\}$ and a set of $M$ center coordinates $\{\mathbf{y}_m\}_{m=1}^M$ with $\mathbf{y}_m \in \mathbb{R}^3$, which the model should predict. We denote predicted node-level labels by $\hat{y}_i \in [0, 1]$ and the set of $K$ predicted center coordinates from the model by $\{\hat{\mathbf{y}}_k\}_{k=1}^K$ with $\hat{\mathbf{y}}_k \in \mathbb{R}^3$.

### 2.2 EGNNs and their Application to Binding Site Identification

EGNNs are straightforward to apply to proteins, when they are represented by a neighborhood graph $\mathcal{P}$, in which each node represents an atom and edges between two atoms represent spatially close atoms (distance between the atoms in the protein is below some threshold). To apply EGNNs, we first set $\mathcal{G} = \mathcal{P}$. For binding pocket identification, one could predict node labels $y_i$, which indicate whether the atom belongs to a binding pocket or not.

The physical nodes represent atoms and their initial coordinate features are set to the location of the atoms $\mathbf{x}_i^0$, and the initial node features $\boldsymbol{h}_i^0$ to, e.g., the atom or residue type. Then, we apply the layer-wise message passing scheme $(\mathbf{X}^{l+1}, \boldsymbol{H}^{l+1}) = \mathrm{EGNN}(\mathbf{X}^l, \boldsymbol{H}^l, \boldsymbol{A})$ (Eqs. (1) to (4)) as given by Satorras et al. (2021):

$$\boldsymbol{m}_{ij} = \boldsymbol{\phi}_e(\boldsymbol{h}_i^l, \boldsymbol{h}_j^l, \|\mathbf{x}_i^l - \mathbf{x}_j^l\|^2, a_{ij}) \tag{1}$$

$$\boldsymbol{m}_i = \sum_{j \in \mathcal{N}(i)} \boldsymbol{m}_{ij} \tag{2}$$

$$\mathbf{x}_i^{l+1} = \mathbf{x}_i^l + \frac{1}{|\mathcal{N}(i)|} \sum_{j \in \mathcal{N}(i)} \frac{\mathbf{x}_i^l - \mathbf{x}_j^l}{\|\mathbf{x}_i^l - \mathbf{x}_j^l\|} \phi_x(\boldsymbol{m}_{ij}) \tag{3}$$

$$\boldsymbol{h}_i^{l+1} = \phi_h(\boldsymbol{h}_i^l, \boldsymbol{m}_i), \tag{4}$$

where $\boldsymbol{\phi}_e$, $\phi_{\mathbf{x}}$ and $\phi_h$ denote multilayer-perceptrons (MLPs). To identify binding pockets, we can extract predictions $\hat{y}_i$ for each atom $i$ by a read-out function applied to the output of the last message passing step $L$, i.e., $\hat{y}_i = \sigma(\boldsymbol{w}^\top \boldsymbol{h}_i^L)$ with an activation function $\sigma$ and parameters $\boldsymbol{w}$. Our model does not incorporate edge features, symbolized by $a_{ij}$. Hence, we will exclude these from the subsequent EGNN formulations and their related derivations.

### 2.3 VN-EGNN: Extension of EGNN with Virtual Nodes

We now extend $\mathcal{G}$ (which is set to the protein neighborhood graph $\mathcal{P}$ for the task of protein binding site identification) with a set of $K$ virtual nodes, which exhibit edges to all other nodes, which will allow us to learn representations of hidden geometric entities, such as binding sites, and simultaneously ameliorate oversquashing. To be able to process this extended graph, we modify EGNNs by locating the virtual nodes at coordinates $\mathbf{Z} = (\mathbf{z}_1, \ldots, \mathbf{z}_K) \in \mathbb{R}^3$ and associating

them with a set of properties $\boldsymbol{V} = (\boldsymbol{v}_1, \ldots, \boldsymbol{v}_K) \in \mathbb{R}^D$. The new message passing scheme $(\boldsymbol{X}^{l+1}, \boldsymbol{H}^{l+1}, \boldsymbol{Z}^{l+1}, \boldsymbol{V}^{l+1}) = $ VN-EGNN $(\boldsymbol{X}^l, \boldsymbol{H}^l, \boldsymbol{Z}^l, \boldsymbol{V}^l)$ of a single VN-EGNN layer consists of three phases (Eqs. (7) to (10), Eqs. (11) to (14), and, Eqs. (15) to (18)), in which the feature and coordinate embeddings of the physical nodes are updated twice:

$$\boldsymbol{h}_i^l \to \boldsymbol{h}_i^{l+1/2} \to \boldsymbol{h}_i^{l+1}, \quad \mathbf{x}_i^l \to \mathbf{x}_i^{l+1/2} \to \mathbf{x}_i^{l+1} \quad \forall i \tag{5}$$

while virtual node embeddings are only updated once per message passing layer

$$\boldsymbol{v}_k^l \to \boldsymbol{v}_k^{l+1}, \quad \boldsymbol{z}_k^l \to \boldsymbol{z}_k^{l+1} \quad \forall k. \tag{6}$$

**Message passing phase I** between **physical nodes** (analogous to EGNN):

$$\boldsymbol{m}_{ij}^{(aa)} = \boldsymbol{\phi}_{e^{(aa)}}(\boldsymbol{h}_i^l, \boldsymbol{h}_j^l, \|\mathbf{x}_i^l - \mathbf{x}_j^l\|) \tag{7}$$

$$\boldsymbol{m}_i^{(aa)} = \sum_{j \in \mathcal{N}(i)} \boldsymbol{m}_{ij}^{(aa)} \tag{8}$$

$$\mathbf{x}_i^{l+1/2} = \mathbf{x}_i^l + \frac{1}{|\mathcal{N}(i)|} \sum_{j \in \mathcal{N}(i)} \frac{\mathbf{x}_i^l - \mathbf{x}_j^l}{\|\mathbf{x}_i^l - \mathbf{x}_j^l\|} \phi_{\mathbf{x}^{aa}}(\boldsymbol{m}_{ij}^{(aa)}) \tag{9}$$

$$\boldsymbol{h}_i^{l+1/2} = \boldsymbol{h}_i^l + \boldsymbol{\phi}_{h^{(aa)}}\left(\boldsymbol{h}_i^l, \boldsymbol{m}_i^{(aa)}\right). \tag{10}$$

**Message passing phase II** from **physical nodes** to **virtual nodes**:

$$\boldsymbol{m}_{ij}^{(av)} = \boldsymbol{\phi}_{e^{(av)}}(\boldsymbol{v}_i^l, \boldsymbol{h}_j^{l+1/2}, \|\boldsymbol{z}_i^l - \mathbf{x}_j^{l+1/2}\|) \tag{11}$$

$$\boldsymbol{m}_i^{(av)} = \frac{1}{N} \sum_{j=1}^{N} \boldsymbol{m}_{ij}^{(av)} \tag{12}$$

$$\boldsymbol{z}_i^{l+1} = \boldsymbol{z}_i^l + \frac{1}{N} \sum_{j=1}^{N} \frac{\boldsymbol{z}_i^l - \mathbf{x}_j^{l+1/2}}{\|\boldsymbol{z}_i^l - \mathbf{x}_j^{l+1/2}\|} \phi_{\mathbf{x}^{av}}(\boldsymbol{m}_{ij}^{(av)}) \tag{13}$$

$$\boldsymbol{v}_i^{l+1} = \boldsymbol{v}_i^l + \boldsymbol{\phi}_{h^{(av)}}\left(\boldsymbol{v}_i^l, \boldsymbol{m}_i^{(av)}\right) \tag{14}$$

**Message passing phase III** from **virtual nodes** to **physical nodes**:

$$\boldsymbol{m}_{ij}^{(va)} = \boldsymbol{\phi}_{e^{(va)}}(\boldsymbol{h}_i^{l+1/2}, \boldsymbol{v}_j^{l+1}, \|\mathbf{x}_i^{l+1/2} - \boldsymbol{z}_j^{l+1}\|) \tag{15}$$

$$\boldsymbol{m}_i^{(va)} = \sum_{j=1}^{K} \boldsymbol{m}_{ij}^{(va)} \tag{16}$$

$$\mathbf{x}_i^{l+1} = \mathbf{x}_i^{l+1/2} + \frac{1}{K} \sum_{j=1}^{K} \frac{\mathbf{x}_i^{l+1/2} - \boldsymbol{z}_j^{l+1}}{\|\mathbf{x}_i^{l+1/2} - \boldsymbol{z}_j^{l+1}\|} \phi_{\mathbf{x}^{va}}(\boldsymbol{m}_{ij}^{(va)}) \tag{17}$$

$$\boldsymbol{h}_i^{l+1} = \boldsymbol{h}_i^{l+1/2} + \boldsymbol{\phi}_{h^{(va)}}\left(\boldsymbol{h}_i^{l+1/2}, \boldsymbol{m}_i^{(va)}\right) \tag{18}$$

Here, $\boldsymbol{\phi}_{e^{(aa)}}, \ldots, \boldsymbol{\phi}_{h^{(va)}}$ are again MLPs. The MLPs $\phi_{\cdot}$ are layer-specific, i.e. $\phi^l$ and currently do not consider edge features. To keep the notation uncluttered, we skipped the layer index $l$ for the MLPs in the formulae above. We call this message passing scheme *heterogeneous* because the different types of messages are generated in subsequent phases.

## 2.4 INITIALIZATION OF VIRTUAL NODES IN VN-EGNN.

It is possibly to equivariantly initialize virtual nodes. To do so, we can initialize the coordinates of the virtual nodes $z_k^0$ at the center of mass, concretely the average coordinates of the physical nodes (Zhang et al., 2024; Kaba et al., 2023), while the initial features $v_k^0$ are learned feature vectors that are pairwise distinct. These choices lead to maintenance of the equivariance properties, but guarantee

that the virtual nodes are updated differently during message passing. Note, that the center of mass is not necessarily a very meaningful value for binding site identification, although it ensures invariant binding site predictions. In practice, this might therefore restrict the architecture quite a lot. In line with new developments such as Abramson et al. (2024), we relaxed the initialization procedure to specifically exploit our prior knowledge, that ligands might tend to bind to surface areas of proteins.

In detail, the relaxed initialization procedure distributes the $K$ virtual nodes evenly across a sphere using a Fibonacci grid (Swinbank and James Purser, 2006), of which the radius is defined as the distance between the protein center and its most distant atom. The virtual node properties $\boldsymbol{v}_k^0$ are initialized by averaging over the initial features $\boldsymbol{h}_i^0$. This procedure is simple and efficient.

**Data augmentation** To prevent virtual nodes from starting at identical locations during different epochs of training, we randomly rotate the sphere with the initial locations of virtual nodes.

## 2.5 PROPERTIES OF VN-EGNN

The following proposition shows, that analogously to EGNNs, VN-EGNNs are equivariant with respect to roto-translations and reflections by construction.

**Proposition 1.** *Equivariant graph neural networks with virtual nodes as defined in Eqs. (7) to (18) are equivariant with respect to roto-translations and reflections of the input and virtual node coordinates.*

*Proof.* See App. E. □

**Virtual nodes ameliorate oversquashing by bounding the maximal shortest-path distance between nodes and required message passing steps.** Several works (Alon and Yahav, 2021; Topping et al., 2022; Di Giovanni et al., 2023) have investigated the relation between oversquashing and characteristics of the MPNN layers and the adjacency matrix. According to Topping et al. (2022) oversquashing is defined as $\frac{\partial h_i^{r+1}}{\partial h_j^0}$, which is the effect that one node with index $j$ has on a node with index $i$ during learning, where the nodes are at a shortest-path distance of $r + 1$. Critically, this quantity can be bounded by the model parameters of the involved MLPs and the topology of the graph (Di Giovanni et al., 2023), concretely the normalized adjacency matrix. We use Topping et al. (2022, Lemma 1), which states that $\left| \frac{\partial h_i^{r+1}}{\partial h_j^0} \right| \leq (\alpha\beta)^{r+1} (\hat{\boldsymbol{A}}^{r+1})_{ij}$ where $\alpha$ and $\beta$ are bounds on the element-wise gradients of the MLPs of the message passing network, $h_i^{r+1}$ is one component of the node representation of node $i$ in message passing layer $r + 1$. The quantity $r + 1$ is both the number of message passing layers and the shortest-path distance between nodes $i$ and $j$ in the graph, and $\hat{\boldsymbol{A}}$ is the normalized adjacency matrix, for which the diagonal values of the original matrix are set to $1$. The normalized adjacency matrix $\hat{\boldsymbol{A}}$ is a symmetric positive matrix that has a leading eigenvalue at $1$ (Perron, 1907; Frobenius, 1912), such that all eigenvalues of all other eigenvectors of $\hat{\boldsymbol{A}}^r$ decay exponentially with $r$. Depending on the weights and activation functions of the MLPs, $|\alpha\beta|^{r+1}$ either grows or vanishes exponentially with $r$, which might lead to either exploding or vanishing gradients, respectively. Thus, learning can only be stabilized via keeping $r$ stable, which virtual nodes that are connected to all other nodes can provide since they bound both the maximal path distance and the necessary number of message passing steps by $r + 1 = 2$.

**Expressiveness of VN-EGNN.** The expressive power of GNNs is linked to their ability to distinguish non-isomorphic graphs. While a minimum of $k$ layers of an EGNN is required to distinguish two $k$-hop distinct graphs, one layer of VN-EGNN is presumed to be sufficient, as can be shown by the application of the Geometric Weisfeiler-Leman test, which serves as an upper bound on the expressiveness of EGNNs. Experimental findings on $k$-chain geometric graphs support this proposition and demonstrate the increased expressive power of VN-EGNN compared to EGNNs without virtual nodes. For further details and an empirical study see App. K.

## 2.6 ADJUSTMENTS OF VN-EGNN FOR BINDING SITE IDENTIFICATION.

**SE(3) equivariance through feature encoding.** We break the equivariance property for mirroring through feature encoding. Since each node has an initial feature that codes for the amino acid, either one-hot encoding or ESM embeddings (Lin et al., 2023), we naturally encode L- and D-amino

acids differently, which leads to different initial features of the initial nodes after mirroring, and consequently breaks E(3) symmetry to SE(3).

## 2.7 TRAINING VN-EGNNS

**Objective.** Previous methods, which consider binding site identification as a node-level prediction task (see Section 2.2) with $\hat{y}_n = \sigma(\boldsymbol{w}^\top \boldsymbol{h}_n^L)$, use a type of *segmentation loss*. The segmentation loss can either be the cross-entropy loss CE:

$$\mathcal{L}_{\text{segm}} = \frac{1}{N} \sum_{n=1}^{N} \text{CE}(y_n, \hat{y}_n)$$

or the Dice loss, that is based on the continuous Dice coefficient (Shamir et al., 2019), with $\epsilon = 1$:

$$\mathcal{L}_{\text{dice}} := 1 - \frac{2 \sum_{n=1}^{N} y_n \, \hat{y}_n + \epsilon}{\sum_{n=1}^{N} y_n + \sum_{n=1}^{N} \hat{y}_n + \epsilon}.$$

The introduction of virtual nodes with coordinates allows to directly tackle the more challenging problem of predicting binding site center points and extracting predictions for these points as outputs of the last EGNN layer. For each protein in the training set, we know the geometric centers of its annotated binding sites, which we denote as $\{\mathbf{y}_1, \ldots, \mathbf{y}_M\}$. The read-out $\hat{\mathbf{y}}_k$ for each virtual node $\hat{\mathbf{y}}_k := \mathbf{z}_k^L$ ($1 \leq k \leq K$) corresponds to its coordinate embedding $\mathbf{z}_k^L$ in the last layer $L$. Each known binding site center should be detected by at least one virtual node, via its read-out, which leads to the following objective

$$\mathcal{L}_{\text{bsc}} = \frac{1}{M} \sum_{m=1}^{M} \min_{k \in 1, \ldots, K} \|\mathbf{y}_m - \hat{\mathbf{y}}_k\|^2. \tag{19}$$

The full objective of VN-EGNN for binding site identification is

$$\mathcal{L} = \mathcal{L}_{\text{bsc}} + \mathcal{L}_{\text{dice}},$$

in which the two terms could also be balanced against each other through a hyperparameter, which we found was not necessary.

**Self-confidence module.** We employ a self-confidence module (Jumper et al., 2021; Zhang et al., 2023a), to assess the quality of predicted binding sites, by equipping each prediction with a confidence score. This allows a ranking of the predictions, similar to Krivák and Hoksza (2018). The confidence value, indicated by $\hat{c}_k$, is computed through $\hat{c}_k = \psi(\boldsymbol{v}_k)$, with $\psi$ implemented as an MLP. During training, the target values for the confidence prediction are generated on-the-fly from the predicted positions $\hat{\mathbf{y}}_k$ and the closest known binding pocket center $\mathbf{y}_m$, in analogy with confidence scores for object detection methods in computer vision.

The confidence label for the $k$-th virtual node is obtained by (Zhang et al., 2023a):

$$c_k = \begin{cases} 1 - \frac{1}{2\gamma} \cdot \|\mathbf{y}_m - \hat{\mathbf{y}}_k\| & \text{if } \|\mathbf{y}_m - \hat{\mathbf{y}}_k\| \leq \gamma, \\ c_0 & \text{otherwise} \end{cases}, \tag{20}$$

with $c_0 = 0.001$. To align with the commonly accepted threshold value of 4Å for the DCC/DCA success rates, we choose $\gamma = 4$. The loss on the confidence score is a mean squared error loss:

$$\mathcal{L}_{\text{confidence}} = \frac{1}{K} \sum_{k=1}^{K} (c_k - \hat{c}_k)^2. \tag{21}$$

## 3 EXPERIMENTS

### 3.1 DATA

We use the benchmarking setting of Zhang et al. (2023b) to perform experiments on four datasets for binding site identification: **scPDB** (Desaphy et al., 2015), **PDBbind** (Wang et al., 2004), **COACH420** and **HOLO4K**. For details, see App. G.1.

Table 1: Performance at binding site identification in terms of DCC and DCA success rates.[a] The first column provides the method, the second the number of parameters of the model, the fourth and the fifth column the performance on the COACH420 dataset, the sixth and seventh column the performance on the HOLO4K dataset, and the remaining columns the performance on PDBbind2020. The best performing method(s) per column are marked bold. The second best in italics.

| Methods | Param | COACH420 | | HOLO4K[d] | | PDBbind2020 | |
|---|---|---|---|---|---|---|---|
| | (M) | DCC↑ | DCA↑ | DCC↑ | DCA↑ | DCC↑ | DCA↑ |
| Fpocket (Le Guilloux et al., 2009)[b] | \ | 0.228 | 0.444 | 0.192 | 0.457 | 0.253 | 0.371 |
| P2Rank (Krivák and Hoksza, 2018)[c] | \ | *0.464* | *0.728* | *0.474* | **0.787** | *0.653* | **0.826** |
| DeepSite (Jiménez et al., 2017)[b] | 1.00 | \ | 0.564 | \ | 0.456 | \ | \ |
| Kalasanty (Stepniewska-Dziubinska et al., 2020)[b] | 70.64 | 0.335 | 0.636 | 0.244 | 0.515 | 0.416 | 0.625 |
| DeepSurf (Mylonas et al., 2021)[b] | 33.06 | 0.386 | 0.658 | 0.289 | 0.635 | 0.510 | 0.708 |
| DeepPocket (Aggarwal et al., 2022b)[c] | \ | 0.399 | 0.645 | 0.456 | 0.734 | 0.644 | 0.813 |
| GAT (Veličković et al., 2018)[b] | **0.03** | 0.039(0.005) | 0.130(0.009) | 0.036(0.003) | 0.110(0.010) | 0.032(0.001) | 0.088(0.011) |
| GCN (Kipf and Welling, 2017)[b] | 0.06 | 0.049(0.001) | 0.139(0.010) | 0.044(0.003) | 0.174(0.003) | 0.018(0.001) | 0.070(0.002) |
| GAT + GCN[b] | 0.08 | 0.036(0.009) | 0.131(0.021) | 0.042(0.003) | 0.152(0.020) | 0.022(0.008) | 0.074(0.007) |
| GCN2 (Chen et al., 2020)[b] | 0.11 | 0.042(0.098) | 0.131(0.017) | 0.051(0.004) | 0.163(0.008) | 0.023(0.007) | 0.089(0.013) |
| SchNet (Schütt et al., 2017)[b] | 0.49 | 0.168(0.019) | 0.444(0.020) | 0.192(0.005) | 0.501(0.004) | 0.263(0.003) | 0.457(0.004) |
| EGNN (Satorras et al., 2021)[b] | 0.41 | 0.156(0.017) | 0.361(0.020) | 0.127(0.005) | 0.406(0.004) | 0.143(0.007) | 0.302(0.006) |
| EquiPocket (Zhang et al., 2023b)[b] | 1.70 | 0.423(0.014) | 0.656(0.007) | 0.337(0.006) | *0.662*(0.007) | 0.545(0.010) | 0.721(0.004) |
| VN-EGNN (ours) | 1.20 | **0.605(0.009)** | **0.750(0.008)** | **0.532(0.021)** | 0.659(0.026) | **0.669(0.015)** | 0.820(0.010) |

[a] The standard deviation across training re-runs is indicated in parentheses.
[b] Results from Zhang et al. (2023b).
[c] Uses different training set and, thus, limited comparability.
[d] This dataset represents a strong domain shift from the training data for all methods (except for P2Rank). Details on the domain shift in App. J.

Table 2: Ablation study. The main components of the VN-EGNN architecture are ablated and tested for their performance on the benchmarking datasets. The first column reports the variant of the ablated method, the second column whether the method contains virtual nodes (VN), the third column whether the method applies heterogenous message passing, and the fourth column whether ESM embeddings were used. The remaining columns are analogous to Table 1.

| Methods | VN | heterog. MP | ESM | COACH420 | | HOLO4K | | PDBbind2020 | |
|---|---|---|---|---|---|---|---|---|---|
| | | | | DCC↑ | DCA↑ | DCC↑ | DCA↑ | DCC↑ | DCA↑ |
| EGNN+VN (Satorras et al., 2021)[b] | ✗ | ✗ | ✗ | 0.156(0.017) | 0.361(0.020) | 0.127(0.005) | 0.406(0.004) | 0.143(0.007) | 0.302(0.006) |
| VN-EGNN (VN only) | ✓ | ✗ | ✗ | 0.497(0.014) | 0.700(0.013) | 0.414(0.023) | 0.618(0.024) | 0.502(0.029) | 0.717(0.025) |
| VN-EGNN (residue emb.) | ✓ | ✓ | ✗ | 0.503(0.022) | 0.684(0.016) | 0.438(0.019) | 0.605(0.013) | 0.551(0.017) | 0.751(0.009) |
| VN-EGNN (homog.) | ✓ | ✗ | ✓ | 0.575(0.008) | 0.708(0.009) | 0.479(0.012) | 0.595(0.010) | 0.649(0.010) | 0.805(0.006) |
| VN-EGNN (full) | ✓ | ✓ | ✓ | **0.605(0.009)** | **0.750(0.008)** | **0.532(0.021)** | **0.659(0.026)** | **0.669(0.015)** | **0.820(0.010)** |

[a] The standard deviation across training re-runs is indicated in parentheses.
[b] Results from Zhang et al. (2023b).

## 3.2 EVALUATION

**Methods compared.** We compare the following binding site identification methods from different categories: *Geometry-based*: Fpocket (Le Guilloux et al., 2009) and P2Rank (Krivák and Hoksza, 2018). *CNN-based*: DeepSite (Jiménez et al., 2017), Kalasanty (Stepniewska-Dziubinska et al., 2020), and DeepSurf (Mylonas et al., 2021). *Topological graph-based*: GAT (Veličković et al., 2018), GCN (Kipf and Welling, 2017), and GCN2 (Chen et al., 2020). *Spatial graph-based*: SchNet (Schütt et al., 2017), EGNN (Satorras et al., 2021), EquiPocket (Zhang et al., 2023b), and our proposed VN-EGNN.

**Evaluation metrics.** We used the *DCC/DCA success rate*, which are well-established metrics for binding site identification (see e.g., Chen et al., 2011). *DCC* is defined as the distance between the predicted and known binding site centers, whereas *DCA* is defined as the shortest distance between the predicted center and any atom of the ligand. Following Stepniewska-Dziubinska et al. (2020) and Zhang et al. (2023b), predictions within a certain threshold of DCC and DCA, are considered as successful, which is commonly referred to as DCC/DCA success rate. Adhering to these works, we maintained a threshold of 4Å throughout our experiments (for other thresholds, see Fig. 2). In line with Chen et al. (2011); Zhang et al. (2023b); Stepniewska-Dziubinska et al. (2020) for each protein only $M$ predicted binding sites with the highest self-confidence scores $\hat{c}_k$ are considered, where $M$ is the number of known binding sites of the protein. Subsequently, each predicted binding site was aligned with the closest real binding site and DCC/DCA success rate was calculated.

## 3.3 IMPLEMENTATION DETAILS.

We used an AdamW (Loshchilov and Hutter, 2019) optimizer for 1500 epochs, selecting the best checkpoint based on the validation dataset. We used 5 VN-EGNN layers, where each layer consists of the three-step message passing scheme described in Section 2.3, and the feature and message size was set to 100, in all layers. Due to the possibility of different virtual nodes converging to identical

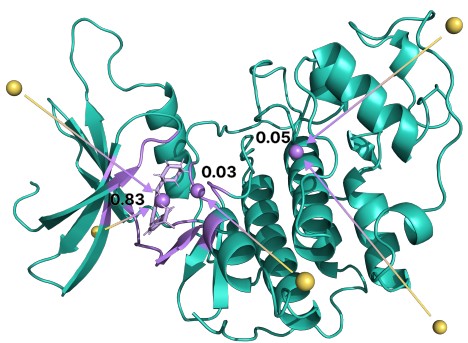 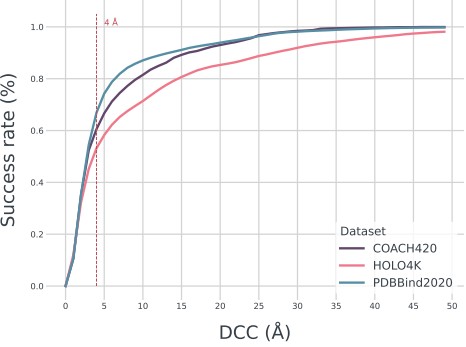

Figure 2: **Left:** Model prediction showing initial positions of the virtual nodes (yellow spheres), ground truth ligand (violet), annotated binding site (violet protein regions), and node position changes (arrows). Violet spheres show clustered virtual node predictions with self-confidence scores. For better visualization, only a subset of initial node positions is shown. (PDB: 1MXI-A) **Right:** DCC success rates at varying thresholds for the distance between predicted and known binding pocket centers in Å.

locations, we employed the Mean Shift Algorithm (Comaniciu and Meer, 2002) at inference time, to cluster virtual nodes that are in close spatial proximity. By averaging their self-confidence scores and positions, we treated these clustered nodes as a single instance. Because of the large complexes in HOLO4K, we ran VN-EGNN for each chain and merged the predicted pocket centers. For the initial residue node features we used pre-trained ESM-2 (Lin et al., 2023) protein embeddings following Corso et al. (2023); Pei et al. (2023). For virtual nodes, we derived their features by averaging the residue node features across the entire protein. We used the position of the $\alpha$-carbons as residue node locations. Virtual nodes are connected to all residue nodes, but not to each other. A linear layer was used to map these initial features to the required dimensions ($\boldsymbol{h}_n^0, \boldsymbol{v}_k^0$) of the model. Layer normalization and Dropout (Srivastava et al., 2014) was applied in each message passing layer. SiLU Hendrycks and Gimpel (2016) activation was used across all layers. Analogous to Pei et al. (2023) we applied normalization (divided by 5) and unnormalization (multiplied by 5) on the coordinates and used the Huber loss (Huber, 1964) for the coordinates, which empirically proved to be slightly more effective. The learning rate was set to $10^{-3}$, after 100 epochs we reduced the learning rate by factor of $10^{-1}$ if the model did not improve for 10 epochs. For training we used 4x NVIDIA A100 40GB GPUs with a batch size of 64 on each GPU. The training time was about 8 hours. Hyperparameters were selected based on a validation dataset which consisted of a 10%-split of the training data (see Table G1).

### 3.4 RESULTS

Our experimental results demonstrate that our method, VN-EGNN, surpasses all prior approaches in terms of the DCC metric on COACH420, PDBbind2020 and even on the challenging HOLO4K dataset, see Table 1. On COACH420, VN-EGNN exhibits the best DCA score and on PDBbind it yields the same DCA score as P2Rank. Note that there is limited comparability with P2rank since this method uses a different training set that might be closer to HOLO4K. HOLO4K contains many complexes of symmetric proteins (see App. G.2), which should be considered as a strong domain shift to the training data and thus pose a problem for all methods, except P2rank. For a more detailed discussion and a visual analysis, we refer to App. J. Visualizations of the predictions of our model are shown in Fig. 2 and Fig. I1. We evaluate predictions of our model with respect to the Dice Loss and to IoU in App. H.3. Memory utilization is shown in Fig. M1 (see App. M).

### 3.5 ABLATION STUDY

Our proposed method comprises three main components as compared to typical other methods: (a) virtual nodes, (b) heterogenous message passing, and (c) pre-trained protein embeddings as node representations. We ablate these three components in a set of experiments (see Table 2). *(a) Removing virtual nodes.* We compared our model to a standard EGNN framework to determine the added value

of virtual nodes. In this study, we analysed how a standard EGNN performs compared to our method. Table 2, shows that the standard EGNN architecture did not perform well. *(b) Homogeneous message passing.* Our approach to message passing, which is applied in a sequential manner, was contrasted with the traditional method where updates across nodes occur in parallel or homogeneously. This evaluation was further enriched by employing identical MLPs for both graph and virtual nodes across all layers, providing a direct comparison of the impact of our message passing strategy. *(c) Atom type embeddings.* We evaluated the impact of the type of embeddings, as outlined in Section 2.2. Table 2 illustrates that, regardless of the initial embeddings used, our model surpasses all preceding approaches, except P2Rank, in achieving higher DCC success rate across the COACH420 and PDBbind2020 datasets. This was accomplished by adopting a one-hot encoding scheme solely for the amino acid types, complemented by an additional category for the virtual nodes.

We provide additional insights on the usage of a different number of virtual nodes and a different number of layers in App. L.

## 4 DISCUSSION AND CONCLUSIONS.

**Main findings.** We have introduced a novel method that extends EGNNs (Satorras et al., 2021) with virtual nodes and a heterogeneous message passing scheme. These new assets improve the learning dynamics by ameliorating the oversquashing problem and enable the learning of representations of hidden geometric entities. Concretely, we have developed this method for binding site identification, for which our experiments show that VN-EGNN exhibits high predictive performance in terms of DCC and DCA and sets a new state-of-the-art on COACH420, HOLO4K and PDBbind2020. We attribute our improvement to the direct prediction of binding site centers, rather than inferring them from the geometric center of segmented areas, a common practice in previous methods. Relying on segmentation can lead to inaccuracies, especially if a single erroneous prediction impacts the calculated center. Overall, VN-EGNN yields highly accurate predictions of binding site centers.

**Comparison with previous work.** In contrast to previous methods (Mylonas et al., 2021; Zhang et al., 2023b), which primarily utilized surface information, based on Sanner et al. (1996); Eisenhaber et al. (1995), or methods that operated on atom-level information (Jiménez et al., 2017; Stepniewska-Dziubinska et al., 2020; Aggarwal et al., 2022b), our approach exclusively relies on residue-level information, specifically using $\alpha$-carbons as physical nodes. This strategy significantly enhances computational efficiency during both training and inference due to the reduced size of the input graphs. Our results support the finding by Jumper et al. (2021), that residue-level information inherently includes all relevant side-chain conformations.

**Limitations.** Our method is currently limited to predicting binding pockets of proteins similar to those in PDB. We expect that VN-EGNNs can also be applied to other physical or geometric problems with hidden geometric entities, such as particle flows, however, their performance in these fields remains to be shown. We have developed VN-EGNN with having the application of binding site identification in mind. Usually in this field, there is a very limited number of training data points and therefore methods taking more prior knowledge into account (e.g., in the design of the network architecture, etc.) could prove beneficial over methods not relying much on this knowledge. With more data points available for training the advantage to take prior knowledge into account may however diminish. Note that our method is not a docking method and thus cannot be used to dock ligands to protein structure. However, our predicted binding sites can be used as proposal regions for other methods, which could lead to improved performance and efficiency for docking methods.

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

APPENDIX

# Contents

# A   IMPLICATIONS AND FUTURE WORK

## A.1   IMPLICATIONS

We expect that our empirical results and our new method re-new the interest in theoretically investigating the effect of virtual nodes on the expressivity and the oversquashing problem of GNNs. Practically, we envision that VN-EGNN will be a useful tool in molecular biology and structure-based drug design, which is regularly used to analyze proteins for potential binding pockets and their druggability. On the long run, this could make the drug development process more time- and cost-efficient.

## A.2   FUTURE WORK

We aim at using VN-EGNN to annotate all proteins in PDB with binding sites, and potentially a subset of the 200 million structures AlphaFold DB, with predicted binding pockets and release this annotated dataset.

## B  NOTATION OVERVIEW

Table B1: Overview of used symbols and notations

| Definition | Symbol/Notation | Type |
|---|---|---|
| number of physical nodes | $N$ | $\mathbb{N}$ |
| number of virtual nodes | $K$ | $\mathbb{N}_0$ |
| number of known binding pockets | $M$ | $\mathbb{N}_0$ |
| dimension of node features | $D$ | $\mathbb{N}$ |
| dimension of messages | $E$ | $\mathbb{N}$ |
| number of message passing layers/steps | $L$ | $\mathbb{N}$ |
| node indices | $i, j, k, n$ | $\{1, ..., K\}$ or $\{1, ..., N\}$ |
| binding pocket index | $m$ | $\{1, ..., M\}$ |
| layer/step index | $l$ | $\{1, ..., L\}$ |
| index set of 10 nearest neighbor atoms | $\mathcal{N}(i)$ | $\{1, ..., N\}^{10}$ |
| physical node coordinates | $\mathbf{x}_i^l$ | $\mathbb{R}^3$ |
| virtual node coordinates | $\mathbf{z}_j^l$ | $\mathbb{R}^3$ |
| physical node feature representation | $\boldsymbol{h}_i^l$ | $\mathbb{R}^D$ |
| virtual node feature representation | $\boldsymbol{v}_j^l$ | $\mathbb{R}^D$ |
| edge feature between atoms | $a_{ij}$ | $\mathbb{R}$ |
| edge feature between atom and virtual node | $d_{ij}$ | $\mathbb{R}$ |
| ground-truth atom label | $y_n$ | $\{0,1\}$ |
| predicted atom label | $\hat{y}_n$ | $[0, 1]$ |
| ground-truth binding site center | $\mathbf{y}_m$ | $\mathbb{R}^3$ |
| prediction of binding site center | $\hat{\mathbf{y}}_k$ | $\mathbb{R}^3$ |
| messages* | $\boldsymbol{m}_{ij}^{(aa)}, \boldsymbol{m}_{ij}^{(av)}, \boldsymbol{m}_{ij}^{(va)}$ | $\mathbb{R}^E$ |
| neural networks for message passing*: | | |
|     message calculation | $\boldsymbol{\phi}_{e^{(aa)}}, \boldsymbol{\phi}_{e^{(av)}}, \boldsymbol{\phi}_{e^{(va)}}$ | $\mathbb{R}^D \times \mathbb{R}^D \times \mathbb{R} \times \mathbb{R} \to \mathbb{R}^E$ |
|     coordinate update | $\phi_{\mathbf{x}^{(aa)}}, \phi_{\mathbf{x}^{(av)}}, \phi_{\mathbf{x}^{(va)}}$ | $\mathbb{R}^E \to \mathbb{R}$ |
|     feature update | $\boldsymbol{\phi}_{h^{(aa)}}, \boldsymbol{\phi}_{h^{(av)}}, \boldsymbol{\phi}_{h^{(va)}}$ | $\mathbb{R}^D \times \mathbb{R}^E \to \mathbb{R}^D$ |
| segmentation loss | $\mathcal{L}_{\text{segm}}$ | $\mathbb{R}^N \times \mathbb{R}^N \to \mathbb{R}$ |
| binding site center loss | $\mathcal{L}_{\text{bsc}}$ | $\mathbb{R}^{3 \times M} \times \mathbb{R}^{3 \times K} \to \mathbb{R}$ |

\* The superscripts (aa), (av) and (va) represent the message passing direction (**atom** to **atom**, **atom** to **virtual node**, **virtual node** to **atom**).

## C  PROBLEM STATEMENT

### C.1  REPRESENTATION OF PROTEINS.

The 3D structure of a protein is usually given by some measurement of its atoms that form the primary amino acid sequence of the protein and the absolute coordinates for the atoms are given as 3D points $\mathbf{x} \in \mathbb{R}^3$. The atoms themselves as well as the amino acids are characterized by their physical, chemical and biological properties. We assume that these properties are summarized by feature vectors $\boldsymbol{h} \in \mathbb{R}^D$, which are located at the atom centers (either of all the atoms or only the ones forming the protein backbone). We formally represent proteins by a neighborhood graph $\mathcal{P} = (\mathcal{P}_N, \mathcal{P}_E)$ with $N$ atom-property pairs, i.e. $\mathcal{P}_N = \{(\mathbf{x}_n, \boldsymbol{h}_n)\}_{n=1}^N$ with $\mathbf{x}_n \in \mathbb{R}^3$ and $\boldsymbol{h}_n \in \mathbb{R}^D$ and a set of directed edges $\mathcal{P}_E$ which consist of atom-property pairs $(i, j)$. Each node $i$ has incoming edges from the 10 nearest nodes $j$ that are closer than 10Å according to the Euclidean distance $\|\mathbf{x}_i - \mathbf{x}_j\|$.

### C.2  REPRESENTATION OF BINDING SITES.

Binding sites are regions around or within proteins, to which ligands can potentially bind. Basically, one can either describe binding sites *explicitly* or *implicitly*. In their explicit representation binding sites would be directly described by the location specifics of the regions, where ligands are located, especially by a region center point. In their implicit representation, binding sites would be described by the atoms of the protein, which surround the ligand. Atoms close to the ligand would be marked as binding site atoms. It might be worth mentioning, that several binding sites per protein are possible.

Formally, for the explicit representation, we describe the (experimentally observed) binding site center points of $M$ distinct binding sites by $\mathbf{y}_m \in \mathbb{R}^3$ with $1 \leq m \leq M$. For the implicit representation, we assign to each protein atom $n$ a label $y_n \in \{0, 1\}$, which is set to 1 if the atom center is within the threshold distance of observed binding ligands, and 0 otherwise.

### C.3  OBJECTIVE.

From an abstract point of view, we want to learn a predictive machine learning model $\mathcal{F}$, parameterized by $\omega$, which maps proteins characterized by the positions of their atoms together with their properties to a binary prediction per atom, whether it might form a binding site and to $K$ 3D coordinates representing binding site region center points:

$$
\mathcal{F}_\omega : \overset{N}{\underset{n=1}{\times}} \left( \underbrace{\mathbb{R}^3 \times \mathbb{R}^D}_{\substack{\text{protein 3D atom} \\ \text{coords with} \\ D\text{-dim features}}} \right) \mapsto \underbrace{[0,1]^N}_{\substack{\text{sem. segm.} \\ \text{protein atoms}}} \times \underbrace{\overset{K}{\underset{k=1}{\times}} \mathbb{R}^3}_{\substack{\text{pos. pred.} \\ \text{virt. nodes}}} \tag{C.1}
$$

$$:= \mathcal{F}_\omega^{\text{segm}} \qquad := \mathcal{F}_\omega^{\text{bsc}}$$

$$\mathcal{F}_\omega ((\mathbf{x}_1, \boldsymbol{h}_1), \ldots, (\mathbf{x}_N, \boldsymbol{h}_N)) = ((\hat{y}_1, \ldots, \hat{y}_N), (\hat{\mathbf{y}}_1, \ldots, \hat{\mathbf{y}}_K))$$
$$\mathcal{F}_\omega^{\text{segm}} ((\mathbf{x}_1, \boldsymbol{h}_1), \ldots, (\mathbf{x}_N, \boldsymbol{h}_N)) := \text{proj}_1 \mathcal{F}_\omega ((\mathbf{x}_1, \boldsymbol{h}_1), \ldots, (\mathbf{x}_N, \boldsymbol{h}_N))$$
$$\mathcal{F}_\omega^{\text{bsc}} ((\mathbf{x}_1, \boldsymbol{h}_1), \ldots, (\mathbf{x}_N, \boldsymbol{h}_N)) := \text{proj}_2 \mathcal{F}_\omega ((\mathbf{x}_1, \boldsymbol{h}_1), \ldots, (\mathbf{x}_N, \boldsymbol{h}_N)),$$

where $\text{proj}_i$ is a projection, that gives the $i$-th component (i.e., prediction of the semantic segmenation part or coordinate predicitons or virtual nodes). Note, that for our predictive model, we use a fixed number $K$ of binding point centers, while indeed $M$ might have been observed for a specific protein.

### C.4  UTILIZED LOSS FUNCTIONS.

**Segmentation loss.** For semantic segmentation (i.e., the prediction of $\mathcal{F}_\omega^{\text{segm}}$), we use a Dice loss, that is based on the continuous Dice coefficient (Shamir et al., 2019), with $\epsilon = 1$:

$$
\mathcal{L}_{\text{dice}} = \text{Dice} ((y_1, \ldots, y_N), (\hat{y}_1, \ldots, \hat{y}_N)) := 1 - \frac{2 \sum_{n=1}^N y_n \hat{y}_n + \epsilon}{\sum_{n=1}^N y_n + \sum_{n=1}^N \hat{y}_n + \epsilon} \tag{C.2}
$$

Perfect predictions lead to a Dice loss of $0$, while perfectly wrong predictions would lead to a Dice of $1$ (in case $\epsilon = 0$ and the denominator is $> 0$).

**Binding site center loss.** For prediction of the binding site region center points (i.e., the prediction of $\mathcal{F}_\omega^{\mathrm{bsc}}$), we use the (squared) Euclidean distance between the set of predicted points and the set of observed ones. More specifically, we assume to be given $M$ observed center points $\{\mathbf{y}_1, \ldots, \mathbf{y}_M\}$. Each of the binding site center points should be detected by at least one of the $K$ outputs from $\mathcal{F}_\omega^{\mathrm{bsc}}$, which translates to using the minimum squared distance to any predicted center point for any of the observed center points:

$$\mathcal{L}_{\mathrm{bsc}} = \mathrm{Dist}\left(\{\mathbf{y}_1, \ldots, \mathbf{y}_M\}, \{\hat{\mathbf{y}}_1, \ldots, \hat{\mathbf{y}}_K\}\right) := \frac{1}{M} \sum_{m=1}^{M} \min_{k \in 1, \ldots, K} \|\mathbf{y}_m - \hat{\mathbf{y}}_k\|^2. \quad (\text{C.3})$$

Our optimization objective is then the sum of the Dice and the Dist loss:

$$\alpha \mathrm{Dice}\left((y_1, \ldots, y_N), (\hat{y}_1, \ldots, \hat{y}_N)\right) + \mathrm{Dist}\left(\{\mathbf{y}_1, \ldots, \mathbf{y}_M\}, \{\hat{\mathbf{y}}_1, \ldots, \hat{\mathbf{y}}_K\}\right) \quad (\text{C.4})$$

with the hyperparameter $\alpha = 1$.

# D   BACKGROUND ON GROUP THEORY AND EQUIVARIANCE

A group in the mathematical sense is a set $G$ along with a binary operation $\circ : G \times G \to G$ with the following properties:

- *Associativity*: The group operation is associative, i.e. $(g \circ h) \circ k = g \circ (h \circ k)$ for all $g, h, k \in G$.
- *Identity:* There exists a unique identity element $e \in G$, such that $e \circ g = g \circ e = g$ for all $g \in G$.
- *Inverse:* For each $g \in G$ there is a unique inverse element $g^{-1} \in G$, such that $g \circ g^{-1} = g^{-1} \circ g = e$.
- *Closure:* For each $g, h \in G$ their combination $g \circ h$ is also an element of $G$.

A group action of group $G$ on a set $X$ is defined as a set of mappings $T_g : X \to X$ which associate each element $g \in G$ with a transformation on $X$, whereby the identity element $e \in G$ leaves $X$ unchanged ($T_e(x) = x \quad \forall x \in X$).

An example is the group of translations $\mathbb{T}$ on $\mathbb{R}^n$ with group action $T_t(x) = \mathbf{x} + \mathbf{t} \quad \forall \mathbf{x}, \mathbf{t} \in \mathbb{R}^n$, which shifts points in $\mathbb{R}^n$ by a vector $\mathbf{t}$.

A function $f : X \to Y$ is equivariant to group $G$ with group action $T_g$ if there exists an equivalent group action $S_g : Y \to Y$ on $G$ such that

$$f(T_g(x)) = S_g(f(x)) \quad \forall x \in X, g \in G.$$

For example, a function $f : \mathbb{R}^n \to \mathbb{R}^n$ is translation-equivariant if a translation of an input vector $\mathbf{x} \in \mathbb{R}^n$ by $\mathbf{t} \in \mathbb{R}^n$ leads to the same transformation of the output vector $f(x) \in \mathbb{R}^n$, i.e. $f(\mathbf{x} + \mathbf{t}) = f(\mathbf{x}) + \mathbf{t}$.

Equivariant graph neural networks (EGNNs) $\psi$ as defined by Satorras et al. (2021) exhibit three types of equivariances:

1. *Translation equivariance:* EGNNs are equivariant to column-wise addition of a vector $\mathbf{t} \in \mathbb{R}^n$ to all points in a point cloud $\mathbf{X} \in \mathbb{R}^{n \times N}$: $\psi(\mathbf{X} + \mathbf{t}) = \psi(\mathbf{X}) + \mathbf{t}$.

2. *Rotation and reflection equivariance*: Rotation or reflection of all points in the point cloud by multiplication with an orthogonal matrix $\mathbf{R} \in \mathbb{R}^{n \times n}$ leads to an equivalent rotation of the output coordinates: $\psi(\mathbf{R}\mathbf{X}) = \mathbf{R}\psi(\mathbf{X})$.

   The group spanning all translations, rotations and reflections in $\mathbb{R}^n$ is called Euclidean group, denoted E($n$), as it preserves Euclidean distances. A proof for E($n$)-equivariance of VN-EGNN can be found in App. E.

3. *Permutation equivariance:* The numbering of elements in a point cloud or graph nodes does not influence the output, i.e. multiplication with a permutation matrix $\boldsymbol{P} \in \mathbb{R}^{N \times N}$ leads to the same permutation of output nodes: $\psi(\mathbf{X}\boldsymbol{P}) = \psi(\mathbf{X})\boldsymbol{P}$. This property holds for message passing graph neural networks in general, as they aggregate and update node information based on local neighborhood structure, regardless of the order in which nodes are presented.

# E    EQUIVARIANCE OF VN-EGNN

In this section we show that the equivariance property of EGNN (Satorras et al., 2021) extends to VN-EGNN, i.e., that rotation and reflection by an orthogonal matrix $\mathbf{R} \in \mathbb{R}^{3\times3}$, and translation by a vector $\mathbf{t} \in \mathbb{R}^3$ of atom and virtual node coordinates leads to an equivalent transformation of output coordinates while leaving node features invariant when applying the message passing steps of VN-EGNN.

**Proposition 1.** *(more formal) E(3) equivariant graph neural networks with virtual nodes (VN-EGNN) as defined by the message passing scheme* $\left(\mathbf{X}^{l+1}, \boldsymbol{H}^{l+1}, \mathbf{Z}^{l+1}, \boldsymbol{V}^{l+1}\right) =$ *VN-EGNN* $\left(\mathbf{X}^l, \boldsymbol{H}^l, \mathbf{Z}^l, \boldsymbol{V}^l, \boldsymbol{A}\right)$ *in Eqs. (7) to (18) are equivariant with respect to reflections and roto-translations of the input and virtual node coordinates, i.e., the following holds (equivariance to reflections and roto-translations):*

$$\left(\mathbf{R}\mathbf{X}^{l+1} + \mathbf{t}, \boldsymbol{H}^{l+1}, \mathbf{R}\mathbf{Z}^{l+1} + \mathbf{t}, \boldsymbol{V}^{l+1}\right) = \text{VN-EGNN}\left(\mathbf{R}\mathbf{X}^l + \mathbf{t}, \boldsymbol{H}^l, \mathbf{R}\mathbf{Z}^l + \mathbf{t}, \boldsymbol{V}^l\right), \quad \text{(E.1)}$$

*where the addition $\mathbf{X}^l + \mathbf{t}$ is defined as column-wise addition of the vector $\mathbf{t}$ to the matrix $\mathbf{X}$.*

*Proof.* We use the notation from Section 2.1 and proceed by tracking the propagation of node roto-translations through the VN-EGNN network. First, we want to show invariance in Eq. (7) in phase I of message passing, equivalently to Satorras et al. (2021), i.e.:

$$\boldsymbol{m}_{ij}^{(aa)} = \boldsymbol{\phi}_{e^{(aa)}}(\boldsymbol{h}_i^l, \boldsymbol{h}_j^l, \|\mathbf{R}\mathbf{x}_i^l + \mathbf{t} - [\mathbf{R}\mathbf{x}_j^l + \mathbf{t}]\|^2) = \boldsymbol{\phi}_{e^{(aa)}}(\boldsymbol{h}_i^l, \boldsymbol{h}_j^l, \|\mathbf{x}_i^l - \mathbf{x}_j^l\|^2) \quad \text{(E.2)}$$

Assuming the initial node features $\boldsymbol{h}_i^0$ do not encode information about the original coordinates $\mathbf{x}_i^0$, it remains to be shown that the Euclidean distance between two nodes is also invariant to translation and rotation:

$$\begin{aligned}
\|\mathbf{R}\mathbf{x}_i^l + \mathbf{t} - [\mathbf{R}\mathbf{x}_j^l + \mathbf{t}]\|^2 &= \|\mathbf{R}\mathbf{x}_i^l - \mathbf{R}\mathbf{x}_j^l\|^2 \\
&= (\mathbf{x}_i^l - \mathbf{x}_j^l)^\top \mathbf{R}^\top \mathbf{R}(\mathbf{x}_i^l - \mathbf{x}_j^l) \\
&= (\mathbf{x}_i^l - \mathbf{x}_j^l)^\top \mathbf{I}(\mathbf{x}_i^l - \mathbf{x}_j^l) \\
&= \|\mathbf{x}_i^l - \mathbf{x}_j^l\|^2
\end{aligned} \quad \text{(E.3)}$$

Consequently, the sum over messages (Eq. (8)) and the feature update function (Eq. (10)), which only uses the summed messages and previous node features as input, are invariant as well, leaving the intermediate output feature representations $\boldsymbol{h}_i^{l+1/2}$ independent of coordinate transformations.

For the remaining equation (Eq. (9)) of phase I the equivariance property can be shown as follows, where Eq. (E.3) is used in the first equality:

$$\begin{aligned}
\mathbf{R}\mathbf{x}_i^l + \mathbf{t} + \frac{1}{\mathcal{N}(i)} \sum_{j \in \mathcal{N}(i)} &\frac{\mathbf{R}\mathbf{x}_i^l + \mathbf{t} - [\mathbf{R}\mathbf{x}_j^l + \mathbf{t}]}{\|\mathbf{R}\mathbf{x}_i^l + \mathbf{t} - [\mathbf{R}\mathbf{x}_j^l + \mathbf{t}]\|} \phi_{\mathbf{x}^{aa}}(\boldsymbol{m}_{ij}^{(aa)}) \\
&= \mathbf{R}\mathbf{x}_i^l + \mathbf{t} + \frac{1}{\mathcal{N}(i)} \sum_{j \in \mathcal{N}(i)} \frac{\mathbf{R}\mathbf{x}_i^l + \mathbf{t} - [\mathbf{R}\mathbf{x}_j^l + \mathbf{t}]}{\|\mathbf{x}_i^l - \mathbf{x}_j^l\|} \phi_{\mathbf{x}^{aa}}(\boldsymbol{m}_{ij}^{(aa)}) \\
&= \mathbf{R}\mathbf{x}_i^l + \mathbf{t} + \frac{1}{|\mathcal{N}(i)|} \mathbf{R} \sum_{j \in \mathcal{N}(i), j \neq i} \frac{\mathbf{x}_i^l - \mathbf{x}_j^l}{\|\mathbf{x}_i^l - \mathbf{x}_j^l\|} \phi_{\mathbf{x}^{aa}}(\boldsymbol{m}_{ij}^{(aa)}) \\
&= \mathbf{R}\left(\mathbf{x}_i^l + \frac{1}{|\mathcal{N}(i)|} \sum_{j \in \mathcal{N}(i), j \neq i} \frac{\mathbf{x}_i^l - \mathbf{x}_j^l}{\|\mathbf{x}_i^l - \mathbf{x}_j^l\|} \phi_{\mathbf{x}^{aa}}(\boldsymbol{m}_{ij}^{(aa)})\right) + \mathbf{t} \\
&= \mathbf{R}\mathbf{x}_i^{l+1/2} + \mathbf{t}
\end{aligned} \quad \text{(E.4)}$$

In phase II of message passing, we input the updated physical node coordinates $\mathbf{R}\mathbf{x}_j^{l+1/2} + \mathbf{t}$ from Eq. (E.4) together with virtual node coordinates $\mathbf{R}\mathbf{z}_i^l + \mathbf{t}$, both subjected to identical rotation and

translation. Invariance of Eqs. (11), (12) and (14) can be deduced similarly to above, using the invariance properties of node features $\boldsymbol{v}_i^l$ and $\boldsymbol{h}_j^{l+1/2}$, and Euclidean distance (Eq. (E.3)):

$$
\begin{aligned}
\boldsymbol{m}_{ij}^{(av)} &= \boldsymbol{\phi}_{e^{(av)}}(\boldsymbol{v}_i^l, \boldsymbol{h}_j^{l+1/2}, \|\mathbf{R}\mathbf{z}_i^l + \mathbf{t} - [\mathbf{R}\mathbf{x}_j^{l+1/2} + \mathbf{t}]\|^2) \\
&= \boldsymbol{\phi}_{e^{(av)}}(\boldsymbol{v}_i^l, \boldsymbol{h}_j^{l+1/2}, \|\mathbf{z}_i^l - \mathbf{x}_j^{l+1/2}\|^2)
\end{aligned}
\tag{E.5}
$$

Thus, the output virtual node features $\boldsymbol{v}_i^{l+1}$ are invariant to roto-translations of node coordinates. Note that reflections are also covered by Eq. (E.4) since the distance of two points does not change under reflection.

Equivariance of output virtual node coordinates $\mathbf{z}_i^{l+1}$ follows analogously to Eq. (E.4):

$$
\mathbf{R}\mathbf{z}_i^l + \mathbf{t} + \frac{1}{N}\sum_{j=1}^{N}\frac{\mathbf{R}\mathbf{z}_i^l + \mathbf{t} - [\mathbf{R}\mathbf{x}_j^{l+1/2} + \mathbf{t}]}{\|\mathbf{R}\mathbf{z}_i^l + \mathbf{t} - [\mathbf{R}\mathbf{x}_j^{l+1/2} + \mathbf{t}]\|}\phi_{\mathbf{x}^{av}}(\boldsymbol{m}_{ij}^{(av)}) = \mathbf{R}\mathbf{z}_i^{l+1} + \mathbf{t}
\tag{E.6}
$$

The same derivations of message invariance

$$
\begin{aligned}
\boldsymbol{m}_{ij}^{(va)} &= \boldsymbol{\phi}_{e^{(va)}}(\boldsymbol{h}_i^{l+1/2}, \boldsymbol{v}_j^{l+1}, \|\mathbf{R} + \mathbf{x}_i^{l+1/2}\mathbf{t} - [\mathbf{R}\mathbf{z}_j^{l+1} + \mathbf{t}]\|^2) \\
&= \boldsymbol{\phi}_{e^{(va)}}(\boldsymbol{h}_i^{l+1/2}, \boldsymbol{v}_j^{l+1}, \|\mathbf{x}_i^{l+1/2} - \mathbf{z}_j^{l+1}\|^2)
\end{aligned}
\tag{E.7}
$$

and coordinate equivariance

$$
\mathbf{R}\mathbf{x}_i^{l+1/2} + \mathbf{t} + \frac{1}{K}\sum_{j=1}^{K}\frac{\mathbf{R}\mathbf{x}_i^{l+1/2} + \mathbf{t} - [\mathbf{R}\mathbf{z}_j^{l+1} + \mathbf{t}]}{\|\mathbf{R}\mathbf{x}_i^{l+1/2} + \mathbf{t} - [\mathbf{R}\mathbf{z}_j^{l+1} + \mathbf{t}]\|}\phi_{\mathbf{x}^{va}}(\boldsymbol{m}_{ij}^{(va)}) = \mathbf{R}\mathbf{x}_i^{l+1} + \mathbf{t}
\tag{E.8}
$$

can be applied to phase III (Eqs. (15) to (18)), proving that invariance of feature representations $\boldsymbol{h}_i^{l+1}$ and equivariance of coordinates $\mathbf{x}_i^{l+1}$ holds true for physical nodes as well, thus, proving Proposition 1.

□

# F PROPERTIES OF VIRTUAL NODE INITIALIZATION

## F.1 INVARIANCE WITH RESPECT TO THE INITIAL COORDINATES OF THE VIRTUAL NODES.

Note that Proposition 1 aims for equivariance with respect to rotations of the physical protein nodes $\mathbf{X}$ and arbitrary, but fixed initialized virtual nodes $\mathbf{Z}$. We further want to have predictions, which are approximately invariant to differently chosen initial virtual node coordinates $\mathbf{Z}^0$. This ultimately leads to predictions that are approximately equivariant with respect to E(3)-transformations of the physical protein nodes. In practice, we distribute the initial virtual node coordinates evenly on a sphere according to an algorithm, which constructs a spherical Fibonacci grid (Swinbank and James Purser, 2006). The algorithm provides spherically distributed grid points, which are fixed at certain locations in the 3D space. In order to achieve invariance with respect to differently chosen initial virtual node coordinates, we randomly rotate this grid of initial virtual node coordinates for each sample in every epoch, i.e. there is variation in the relative alignment of the Fibonacci grid points, that represent the virtual node positions, to physical protein nodes. Empirically, we observe that this training strategy leads to approximate invariance to different initializations of the virtual node coordinates (see Table F1).

## F.2 A POTENTIAL ALTERNATIVE STRATEGY FOR INITIALIZATION VIRTUAL NODE COORDINATES.

An idea to avoid random alignments between physical node coordinates and initial virtual node coordinates, would be, to change initial coordinates in an equivariant way with respect to E(3) group transformations of the protein physical nodes. This could be achieved, e.g., by defining frames (Puny et al., 2022) based on Principal Component Analysis of physical protein node coordinates and by aligning the Fibonacci grid relative to these frames. Consequently, we would achieve that binding pocket predictions would change equivariantly with E(3)- transformations of the protein. Thereby the definition of such frames via Principal Component Analysis (PCA) is possible up to certain degenerate cases, that occur with probability zero for proteins. Since the orientation of axes might still not be unique, a strategy might be to compute properties such as the overall molecular weight for each octant in the coordinate system spanned by PCA eigenvectors. The orientation can then be set, such that for the octant with the maximum overall molecular weight, all coordinates get positive values.

Table F1: Mean Performance at binding site identification in terms of DCC and DCA success rates together with their standard deviations (in parentheses). Means and standard deviations are across different random rotations of the Fibonacci grid.

| Dataset | DCC | DCA |
|---|---|---|
| COACH420 | 0.612(0.005) | 0.741(0.006) |
| HOLO4K | 0.524(0.002) | 0.632(0.002) |
| PDBbind2020 | 0.702(0.001) | 0.833(0.002) |

# G    EXPERIMENTAL SETTINGS

## G.1    DATASETS

**scPDB** (Desaphy et al., 2015) is a frequently utilized dataset for binding site prediction (Kandel et al., 2021; Stepniewska-Dziubinska et al., 2020), encompassing both protein and ligand structures. We employed the 2017 release of scPDB in the training and validation. This release comprises 17,594 structures, 16,034 entries, 4,782 proteins, and 6,326 ligands. Structures were clustered based on their Uniprot IDs. From each cluster, protein structures with the longest sequences were selected, in alignment to the strategies used in Kandel et al. (2021) and Zhang et al. (2023b).

(Source: `https://github.com/jivankandel/PUResNet/blob/main/scpdb_subset.zip`)

**PDBbind** (Wang et al., 2004) is a widely recognized dataset integral to the study of protein-ligand interactions. This dataset provides detailed 3D structural information of proteins, ligands, and their respective binding sites, complemented by rigorously determined binding affinity values derived from laboratory evaluations. For our work, we draw upon the v2020 edition, which is divided into two sets: the general set (comprising 14,127 complexes) and the refined set (containing 5,316 complexes). While the general set encompasses all protein-ligand interactions, only the refined set, curated for its superior quality from the general collection, is used in our experiments.

(Source: `http://www.pdbbind.org.cn/download/PDBbind_v2020_refined.tar.gz`)

**COACH420** and **HOLO4K** are benchmark datasets utilized for the prediction of binding sites, as originally detailed by Krivák and Hoksza (2018). Following the methodologies of Krivák and Hoksza (2018); Mylonas et al. (2021); Aggarwal et al. (2022a), we adopt the so-called `mlig` subsets from each of these datasets, which encompass the significant ligands pertinent to binding site prediction. Note that the HOLO4K contains many multi-chain systems and complexes with multiple copies of the protein (see Section App. J), such that this dataset's distribution is strongly differs from the other datasets.

(Source: `https://github.com/rdk/p2rank-datasets`)

For comprehensive data preparation across all datasets, solvent atoms were excluded and erroneous structures were removed.

## G.2    HYPERPARAMETERS AND HYPERPARAMETER SELECTION

Table G1 shows the evaluated hyperparameters. Bold indicates the parameters used in final model.

Table G1: A list of considered and selected hyperparameters.

| hyperparmater | considered and **selected** values |
|---|---|
| optimizer | {**AdamW**, Adam } |
| learning rate | $\{\mathbf{0.001}, 0.0001\}$ |
| activation function | { **SiLU**, ReLU } |
| dimension of node features $D$ | $\{20, 30, \mathbf{100}\}$ |
| dimension of the messages $P$ | $\{40, 50, \mathbf{100}\}$ |
| number of message passing layers/steps $L$ | $\{2, 3, 4, \mathbf{5}\}$ |
| number of virtual nodes $K$ | $\{4, \mathbf{8}\}$ |
| Huber loss $\delta$ | $\{1\}$ |

# H ADDITIONAL INSIGHTS

## H.1 INITIAL EXPERIMENT

Fig. H1 shows the training curves for a VN-EGNN during the development phase. VN-EGNN were only trained to minimize the segmentation loss $\mathcal{L}_{\mathrm{segm}}$. Even in the absence of a the binding site center loss $\mathcal{L}_{\mathrm{bsc}}$, the virtual nodes tend to converge towards the actual binding site center. This finding inspired us to further refine the position of the virtual nodes by including it directly to the optimization objective, which further improved the results significantly.

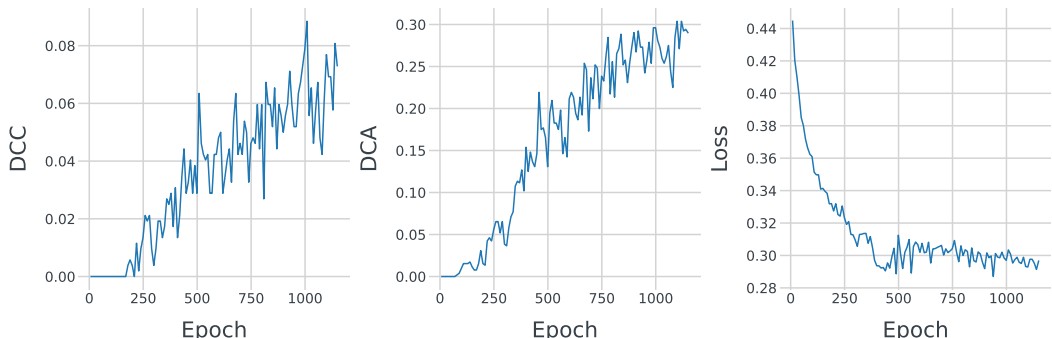

Figure H1: Validation curves of a VN-EGNN during development. Despite only being trained to minimize the segmentation loss, the virtual nodes converged towards the known binding sites. **Left**: DCC success rate during training. **Middle**: DCA success rate during training. **Right**: Segmentation loss during training.

## H.2 VIRTUAL NODE INITIALIZATION STRATEGIES

Fig. H2 shows learning curves for DCC and DCA on the validation set for a strict equivariant initialization strategy (i.e., center of mass (CoM) based initialization (Zhang et al., 2024; Kaba et al., 2023)) and for a relaxed version of this initialization strategy (i.e, Fibonacci grid based initialization, where the grid is rotated during training). For CoM based initialization the node embeddings are learned, whereas for the Fibonacci grid based initialization the virtual node features are initialized by the average of the physical node ESM embeddings of the protein. Our findings suggest that relaxing strict equivariance could be beneficial for binding site identification.

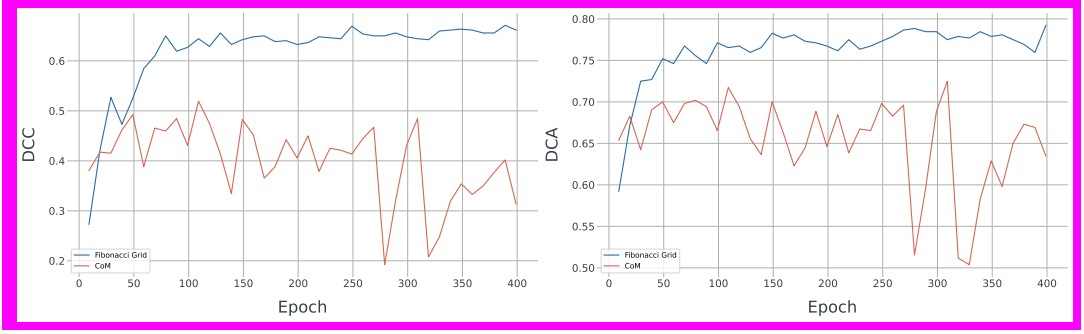

Figure H2: Exemplary Validation set learning curves for strict equivariant CoM based initialization and for Fibonacci grid based initialization. **Left**: DCC metric. **Right**: DCA metric.

### H.3 SEGMENTATION LOSS EVALUATION

In Table H1, we evaluate predictions for the Dice loss, which is a part of our objective and for the Intersection over Union (IoU, see Eq. (H.1)), which is related to Dice loss.

$$\text{IoU}\left((y_1,\ldots,y_N),(\hat{y}_1,\ldots,\hat{y}_N)\right) := \frac{\sum_{n=1}^{N} y_n \, \mathbf{1}_{[\hat{y}_n>0.5]}}{\sum_{n=1}^{N} y_n + \sum_{n=1}^{N} \mathbf{1}_{[\hat{y}_n>0.5]} - \sum_{n=1}^{N} y_n \, \mathbf{1}_{[\hat{y}_n>0.5]}} \tag{H.1}$$

|            | COACH420      | HOLO4K        | PDBbind2020   |
|------------|---------------|---------------|---------------|
| Dice loss  | 0.397(0.015)  | 0.584(0.031)  | 0.357(0.010)  |
| IoU        | 0.437(0.005)  | 0.263(0.025)  | 0.477(0.003)  |

Table H1: Dice and Intersection over Union (IoU) Loss

# I    VISUALIZATIONS

Fig. 2 left and Fig. I1 show exemplary model predictions visualized with Pymol.

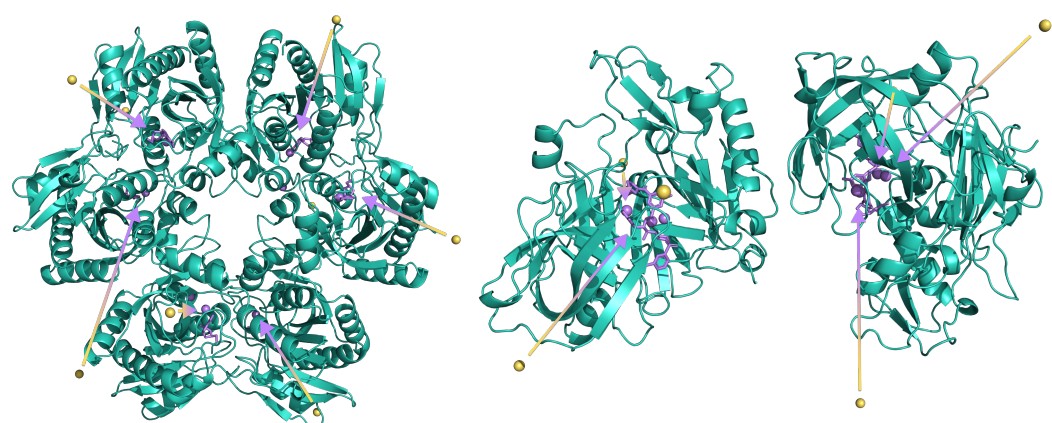

Figure I1: Examples of Detected Binding Sites: Visualization and Analysis. We visualized two distinct proteins using Pymol, where the initial positions of the virtual nodes are represented by yellow spheres, were the violet spheres indicate the virtual nodes following $L$ message passing steps. The violet molecules indicate the position of the ligand as in the original PDB file. The arrows indicate the starting positions and the predicted positions of the virtual nodes. The visualization demonstrate that our model distributes the virtual nodes amongst various possible binding positions. The visualizations show the predicted positions after applying clustering as described in Section 3.3. To simplify the visualization, not all initial positions of the virtual nodes are depicted in the figure. **Left**: PDB: 1ODI **Right**: PDB: 3LPK.

Further, we visualize the learned virtual node features, grouped by the corresponding protein's target classification according to the ChEMBL (Gaulton et al., 2011) database to analyze whether these representations contain relevant information about the protein/binding pocket (Fig. I2).

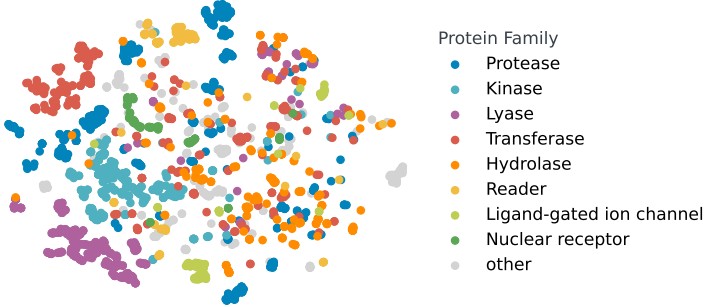

Figure I2: T-SNE embeddings of virtual node features of the best ranked pockets for each protein in the PDBbind2020 dataset colored by protein family according to ChEMBL. The eight largest protein classes are shown, remaining proteins are colored in grey.

# J  DOMAIN SHIFT OF THE HOLO4K DATASET

The HOLO4K benchmark comprises a large set of protein complexes and their annotated binding sites. HOLO4K has often been used as a benchmarking dataset (Krivák and Hoksza, 2018), while it exhibits different characteristics than other datasets, such as scPDB, COACH420 and PDBBind2020. The number of chains per sample, i.e. PDB file, is larger than in these datasets (see Fig. J1), and also the number of binding sites per entry is higher (see Fig. J1). HOLO4K contains many symmetric units of large complexes which lead to these statistics. Thus, for machine learning methods trained on scPDB, the HOLO4K dataset represents a difficult test case due to the mentioned domain shifts.

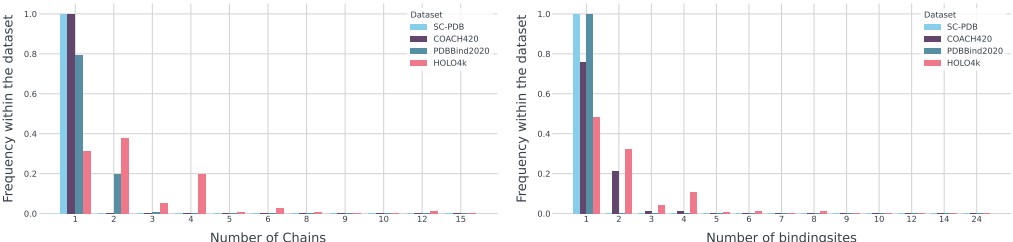

Figure J1: Histogram of the number of protein chains (left) and of the number of binding sites frequency (right) per sample for each of the datasets used in this study. Note that the HOLO4K dataset has highly different characteristics compared to the other datasets and thus represents a strong domain shift for methods trained on scPDB.

## K  EXPRESSIVENESS OF VN-EGNN

The expressive power of GNNs is often described in terms of their ability to distinguish non-isomorphic graphs. The Weisfeiler-Leman (WL) (Weisfeiler and Leman, 1968) test, an iterative method to determine whether two attributed graphs are isomorphic, provides an upper bound to the expressiveness of GNNs. To extend the applicability of this framework to geometric graphs, Joshi et al. (2023) introduced the Geometric Weisfeiler-Leman test (GWL) which assesses whether two graphs are *geometrically isomorphic*.

**Definitions** (Joshi et al., 2023): Two graphs $\mathcal{G}_1$ and $\mathcal{G}_2$ with node features $h_i^{\mathcal{G}_j}$ and coordinates $\mathbf{x}_i^{\mathcal{G}_j}$ for $j \in \{1, 2\}$ are called *geometrically isomorphic* if there exists an edge-preserving bijection $b : \mathcal{V}(\mathcal{G}_1) \to \mathcal{V}(\mathcal{G}_2)$ between their corresponding node indices $\mathcal{V}(\mathcal{G}_j)$, such that their geometric features are equivalent up to $E(n)$ group actions, i.e. global rotations/reflections $\boldsymbol{R}$ and translations $\mathbf{t}$:

$$\left( h_{b(i)}^{\mathcal{G}_2}, \mathbf{x}_{b(i)}^{\mathcal{G}_2} \right) = \left( h_i^{\mathcal{G}_1}, \boldsymbol{R}\mathbf{x}_i^{\mathcal{G}_1} + \mathbf{t} \right) \quad \forall i \in \mathcal{V}(\mathcal{G}_1). \tag{K.1}$$

Two graphs $\mathcal{G}_1$ and $\mathcal{G}_2$ are called *k-hop distinct* if for all graph isomorphisms $b$, there is some node $i \in \mathcal{V}(\mathcal{G}_1), b(i) \in \mathcal{V}(\mathcal{G}_2)$ such that the corresponding $k$-hop neighborhood subgraphs $\mathcal{N}_i^{(\mathcal{G}_1, k)}$ and $\mathcal{N}_{b(i)}^{(\mathcal{G}_2, k)}$ are distinct. Otherwise, if $\mathcal{N}_i^{(\mathcal{G}_1, k)}$ and $\mathcal{N}_{b(i)}^{(\mathcal{G}_2, k)}$ are identical up to group actions for all $i \in \mathcal{V}(\mathcal{G}_1)$, we say $\mathcal{G}_1$ and $\mathcal{G}_2$ are *k-hop identical*.

In addition to iteratively updating node colors depending on node features in the local neighborhood analogously to the WL test, GWL keeps track of $E(n)$-equivariant hash values of each node's local geometry, i.e., distances to and angles between neighboring nodes. Thus, $k$ iterations of GWL are necessary and sufficient to distinguish any $k$-hop distinct, $(k-1)$-hop identical geometric graphs (Joshi et al., 2023).

**Proposition 2.** *Any two geometrically distinct graphs $\mathcal{G}_1$ and $\mathcal{G}_2$, where the underlying attributed graphs are isomorphic, can be distinguished with one iteration of GWL by adding one virtual node connected to all other nodes.*

*Proof.* For 1-hop distinct graphs one iteration of GWL suffices to distinguish them even without virtual nodes and, thus, the proposition holds.

Now, we assume that $\mathcal{G}_1$ and $\mathcal{G}_2$ are $k$-hop distinct and $(k-1)$-hop identical for any $k > 1 \in \mathbb{N}$ and place one virtual node connected to all other nodes in an equivalent position in both graphs.

Note that finding equivalent virtual node positions is not trivial, as there is no straightforward way to spatially align the two graphs. Since the graphs are $(k-1)$-hop identical there is at least one bijective mapping between $(k-1)$-hop sub-graphs of $\mathcal{G}_1$ and $\mathcal{G}_2$, such that the neighborhood structure between matching sub-graphs is preserved.[1] For each such mapping between sub-graphs, we align the two graphs in space by overlaying one matching pair of $(k-1)$-hop sub-graphs (consisting of more than two nodes that are not arranged in a straight line) and position the virtual node in the same coordinates in both aligned graphs.

Since the virtual node is connected to each node in the graph, its 1-hop neighborhood and therefore the receptive field of the first GWL iteration contains the entire graph. Due to the $k$-hop distinctness of the graphs, there exists at least one node for which the geometric orientation relative to the matched subgraph deviates between $\mathcal{G}_1$ and $\mathcal{G}_2$. Thus, the hash values corresponding to the virtual nodes' geometric information differ, and the graphs can be distinguished by only one iteration of GWL. Note that in case there are multiple possible mappings between $(k-1)$-hop subgraphs, $\mathcal{G}_1$ and $\mathcal{G}_2$ have to be distinguishable by one GWL iteration for each such mapping, in order to be classified as geometrically distinct. $\square$

---

[1]In practice, finding such a mapping is non-trivial and can be computationally expensive (Widdowson and Kurlin, 2023).

As $k$ iterations of GWL act as an upper bound on the expressiveness of a $k$-layer geometric GNN, we propose that one layer of VN-EGNN is sufficient to distinguish two $k$-hop distinct graphs while without virtual nodes $k$ EGNN layers are necessary to complete the same task.

We demonstrate this on the example of $n$-chain geometric graphs, where each pair of graphs comprises $n$ nodes arranged in a line and two end points with distinct orientations (Fig. K1). These graphs are $(\lfloor \frac{n}{2} \rfloor + 1)$-hop distinct and should therefore be distinguishable by $(\lfloor \frac{n}{2} \rfloor + 1)$ EGNN layers or $(\lfloor \frac{n}{2} \rfloor + 1)$ iterations of GWL.

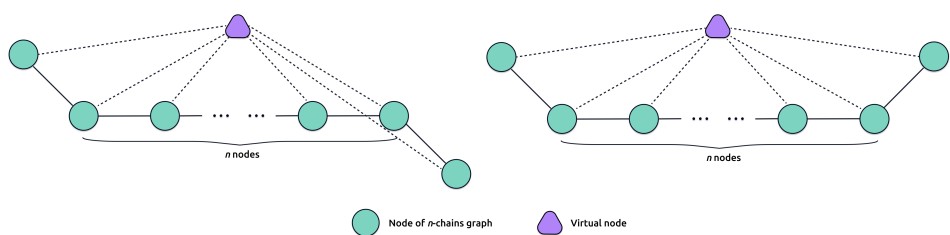

Figure K1: A pair of $n$-chain geometric graphs consisting of $n$ nodes arranged in a line and two end points with opposite orientations. Without the addition of a virtual node, these graphs are $(\lfloor \frac{n}{2} \rfloor + 1)$-hop distinct.

We trained EGNNs with an increasing number of layers to classify 4-chain graphs, both with and without the addition of a virtual node. We tested two different strategies for the virtual node's initial position. Firstly, the virtual node was placed at the same relative position in both graphs, such that when the first $(n + 1)$ nodes of the two $n$-chain graphs are overlaid, the virtual nodes coincide. The exact position was randomly selected on a sphere centered at the midpoint of the $n$-chain. In the second setting, the virtual node was initialized at the center of mass (Zhang et al., 2024; Kaba et al., 2023) of each graph, ensuring equivariance with respect to the initialization. Note that for the experiments including virtual nodes, we did not use the heterogeneous message passing scheme as described in Section 2.3, but apply the EGNN to the entire graph, including virtual nodes, at once.

The results shown in Table K1 demonstrate that, as expected, 3 layers of EGNN are necessary to distinguish the 4-chain graphs while after adding a virtual node, one iteration is sufficient for correct classification, indicating the increased expressiveness of VN-EGNN. Although we used the setting of Joshi et al. (2023), we could not reproduce their finding that 6 EGNN layers are necessary to solve this task, which they explained with possible oversmoothing or oversquashing effects. The differences might arise from the use of different features dimensions, which is why we include results for 5 different feature dimensions.

Table K1: Classification accuracy of EGNNs with and without virtual nodes and increasing node embedding dimensions on 4-chain geometric graphs. The standard deviation across 100 training re-runs is indicated with $\pm$ and column "Dim." indicates the used node feature dimension. Note that VN-EGNN can distinguish these graphs already with one message passing layer (see columns "1 Layer" and "2 Layers").

| | Dim. | 1 Layer | 2 Layers | 3 Layers | 4 Layers | 5 Layers | 6 Layers | 7 Layers | 8 Layers |
|---|---|---|---|---|---|---|---|---|---|
| | 8 | $50.0 \pm 0.0$ | $50.0 \pm 0.0$ | $50.0 \pm 0.0$ | $98.0 \pm 9.8$ | $94.0 \pm 16.2$ | $93.0 \pm 17.3$ | $99.5 \pm 5.0$ | $99.5 \pm 5.0$ |
| | 16 | $50.0 \pm 0.0$ | $50.0 \pm 0.0$ | $86.0 \pm 22.4$ | $97.5 \pm 10.9$ | $99.5 \pm 5.0$ | $99.5 \pm 5.0$ | $99.5 \pm 5.0$ | $100.0 \pm 0.0$ |
| **EGNN** | 32 | $50.0 \pm 0.0$ | $50.0 \pm 0.0$ | $56.5 \pm 16.8$ | $50.0 \pm 0.0$ | $50.0 \pm 0.0$ | $96.5 \pm 12.8$ | $99.0 \pm 7.0$ | $93.5 \pm 16.8$ |
| | 64 | $50.0 \pm 0.0$ | $50.0 \pm 0.0$ | $100.0 \pm 0.0$ | $99.0 \pm 7.0$ | $100.0 \pm 0.0$ | $99.0 \pm 7.0$ | $100.0 \pm 0.0$ | $100.0 \pm 0.0$ |
| | 128 | $50.0 \pm 0.0$ | $50.0 \pm 0.0$ | $96.5 \pm 12.8$ | $98.5 \pm 8.5$ | $95.0 \pm 15.0$ | $99.5 \pm 5.0$ | $99.5 \pm 5.0$ | $99.5 \pm 5.0$ |
| | 8 | $65.5 \pm 23.1$ | $50.0 \pm 0.0$ | $84.5 \pm 23.1$ | $92.5 \pm 17.9$ | $64.0 \pm 22.4$ | $97.0 \pm 11.9$ | $86.5 \pm 23.3$ | $97.5 \pm 10.9$ |
| **VN-EGNN** | 16 | $86.0 \pm 23.5$ | $95.0 \pm 15.0$ | $98.5 \pm 8.5$ | $99.5 \pm 5.0$ | $99.5 \pm 5.0$ | $98.0 \pm 9.8$ | $99.5 \pm 5.0$ | $100.0 \pm 0.0$ |
| (initialization | 32 | $95.0 \pm 15.0$ | $100.0 \pm 0.0$ | $99.5 \pm 5.0$ | $99.5 \pm 5.0$ | $100.0 \pm 0.0$ | $100.0 \pm 0.0$ | $100.0 \pm 0.0$ | $100.0 \pm 0.0$ |
| on a sphere) | 64 | $97.5 \pm 10.9$ | $100.0 \pm 0.0$ | $99.5 \pm 5.0$ | $99.5 \pm 5.0$ | $99.5 \pm 5.0$ | $100.0 \pm 0.0$ | $100.0 \pm 0.0$ | $99.5 \pm 5.0$ |
| | 128 | $99.0 \pm 7.0$ | $99.5 \pm 5.0$ | $99.5 \pm 5.0$ | $99.0 \pm 7.0$ | $99.5 \pm 5.0$ | $99.5 \pm 5.0$ | $99.0 \pm 7.0$ | $99.0 \pm 7.0$ |
| | 8 | $95.0 \pm 15.0$ | $89.0 \pm 20.7$ | $99.5 \pm 5.0$ | $97.5 \pm 10.9$ | $100.0 \pm 0.0$ | $99.5 \pm 5.0$ | $98.5 \pm 8.5$ | $100.0 \pm 0.0$ |
| **VN-EGNN** | 16 | $98.0 \pm 9.8$ | $97.0 \pm 11.9$ | $99.5 \pm 5.0$ | $100.0 \pm 0.0$ | $100.0 \pm 0.0$ | $100.0 \pm 0.0$ | $100.0 \pm 0.0$ | $100.0 \pm 0.0$ |
| (with center of | 32 | $100.0 \pm 0.0$ | $100.0 \pm 0.0$ | $100.0 \pm 0.0$ | $100.0 \pm 0.0$ | $99.5 \pm 5.0$ | $100.0 \pm 0.0$ | $100.0 \pm 0.0$ | $100.0 \pm 0.0$ |
| mass initialization) | 64 | $100.0 \pm 0.0$ | $100.0 \pm 0.0$ | $100.0 \pm 0.0$ | $100.0 \pm 0.0$ | $100.0 \pm 0.0$ | $100.0 \pm 0.0$ | $100.0 \pm 0.0$ | $100.0 \pm 0.0$ |
| | 128 | $99.0 \pm 7.0$ | $99.5 \pm 5.0$ | $100.0 \pm 0.0$ | $99.5 \pm 5.0$ | $100.0 \pm 0.0$ | $100.0 \pm 0.0$ | $100.0 \pm 0.0$ | $100.0 \pm 0.0$ |

## L  ADDITIONAL HYPERPARAMETER EVALUATION

In Table L1 we benchmark VN-EGNN for different numbers of virtual nodes and different numbers of layers than in the main part of this paper. In many cases we still obtain decent results.

| | COACH420 | | HOLO4K | | PDBbind2020 | |
|---|---|---|---|---|---|---|
| | DCC | DCA | DCC | DCA | DCC | DCA |
| 10 virtual nodes | 0.591(0.010) | 0.736(0.010) | 0.530(0.011) | 0.649(0.010) | 0.677(0.013) | 0.813(0.012) |
| 12 virtual nodes | 0.609(0.011) | 0.738(0.020) | 0.521(0.019) | 0.642(0.017) | 0.677(0.010) | 0.825(0.010) |
| 4 layer | 0.595(0.008) | 0.740(0.012) | 0.530(0.019) | 0.646(0.023) | 0.685(0.013) | 0.827(0.008) |
| 6 layer | 0.598(0.017) | 0.731(0.001) | 0.507(0.016) | 0.614(0.029) | 0.673(0.024) | 0.816(0.013) |
| Default | 0.605(0.009) | 0.750(0.008) | 0.532(0.021) | 0.659(0.026) | 0.669(0.015) | 0.820(0.010) |

Table L1: DCC/DCA performance for varying numbers of virtual nodes and message passing layers. All other parameters were maintained as specified in the original paper.

# M    MEMORY UTILIZATION

Since the number of virtual nodes are connected to all other nodes, we show how much memory utilization increases when increasing the number of virtual nodes in Fig. M1. Thereby we consider a range for the number of virtual nodes, which seems to be practically relevant for binding site identification (i.e., up to 16 virtual nodes).

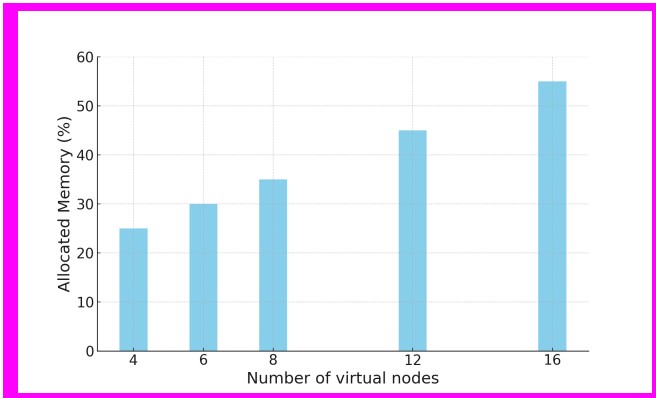

Figure M1: Memory utilization for varying numbers of virtual nodes was assessed as a percentage of total capacity on an NVIDIA A100 GPU (80GB).

