# OpenReview forum: "VN-EGNN: E(3)- and SE(3)-Equivariant Graph Neural Networks with Virtual Nodes Enhance Protein Binding Site Identification"
_ICLR.cc/2025/Conference — Submitted to ICLR 2025_

### Official Review · Reviewer_Xvjk · 2024-10-28

**Soundness:** 3
**Presentation:** 3
**Contribution:** 3
**Rating:** 8
**Confidence:** 3

**Summary:**

In this paper, the authors present a state-of-the-art model for protein binding site identification based on the addition of virtual nodes to the usual EGNN.
Those virtual nodes serve multiple purposes:
- They alleviate some problems recurrent with GNN like smoothing, vanishing, or exploding gradient.
- They encode in their positions the central position of the identified binding site and in their final hidden representation some overall general information about the protein they were added on (like for example protein family). The final virtual nodes' hidden representation also contains some information (by training) about the confidence in those virtual nodes' ability to recover a binding site.
- In the case of graphs with nodes coordinates, those virtual nodes allow for more expressiveness of the network.

**Strengths:**

Particularly, and independently of its state of the art, this paper's novelty and interest lays on:
- An extension of EGNN for graphs with virtual nodes as well as a proof of their power for solving the binding identification task, thanks to an ablation study.
- A new 3-step message passing proven to be powerful compared to its homogeneous counterpart via an ablation study.
- A method to ensure at least approximate virtual node initial positioning invariance (data augmentation), and a discussion on potential ways to improve upon it.
Bonuses:
- The model is equipped with a self-confidence module, which is always useful at prediction time in real-case scenarios.
- The paper offers plenty of discussions about VN-EGNN vs EGNN even outside of the particular task of binding site identification.

**Weaknesses:**

The paper is pretty solid and the few questions that I think would strengthen the paper even more are listed below. They mainly have to do with showcasing more the use of the model at prediction time, clearly demonstrating strength , weakness and protocol associated to using the model. Some part are missing in that area: not sure how easy it would be for someone interested to use this model, to do so. Again mainly at prediction time, because I think the training is well described.

**Questions:**

Suggestions for improvement (suggestions are minors):
- Discussion about the reliability of the self-confidence score (most important comment for me): what typically is the range it outputs, what would be in practice still considered good confidence, how to work with multiple of those, and so on at prediction time. I am pointing this out because of 2 things. First, we only have metrics for DCC and DCA and never talk again about the confidence score. Second, the training loss consists of matching a known binding site to the nearest virtual node. Is it at training time that the virtual nodes clustering is done? Or is this a way to handle unassigned virtual nodes, or binding sites to virtual node multiplicity, at prediction time? I think a bit more details about how to handle those cases with examples, would be super helpful in the appendix.
- Overall a few sentences about the scaling in memory and computation according to protein size could be interesting, as well as quantitative limits in terms of number of binding sites and protein size. The case of  Holo4K is mentioned as a particularly hard case (even though their model still performs there), and even though the training is done on cases with way fewer binding sites, the model can predict up to 8 independent binding sites. Can we have a breaking of model performance by number of binding sites?
- Finally, do the authors have any ideas why they still needed 5 VN-EGNN  layers despite the huge theoretical and experimental (appendix section on expressiveness table) boost in expressiveness given by the inclusion of virtual nodes (and its decoupling to receptive field)? Here the virtual node addition should in principle give you the optimal receptive fields so adding layers should not matter much, except if in that case the model also needs to understand more geometry via N body correlation which is also offered by stacking 2 body message passing layers. But in that case and given what MACE showed I was more expecting 3 to 4 layers to at least have access to angles information. 5 seems a lot from an outsider's perspective.

---

> ### Author Response · Authors · 2024-11-23
>
> We thank the reviewer for acknowledging our efforts to advance the state of the art in methods for protein binding site identification. We thank the reviewer for stating our main contributions so clearly and hope that we can answer their remaining questions in a satisfying way.
>
> #### Questions
>
> ##### Question 1: Confidence Score's Role in DCC/DCA Metrics ...
>
> > With our architecture, we obtain **two** output values for each virtual node: a confidence score and a predicted binding site location. Since, the DCC/DCA metric computation require as many predictions as experimentially found binding sites, we provide the site location predictions from the highest-scoring virtual nodes for computation of DCC/DCA. To achieve high DCC/DCA values, both the prediction of the binding site location and the prediction of the confidence score must work well. Conversely, it applies, that in case confidence predictions would not work well, DCC/DCA would be low. A further aspect concerning the confidence score with respect to potentially weak signals is, that even if lower-confidence predictions would accurately identify binding sites correctly, they are excluded from the metric calculation. In light of having obtained new state-of-the-art results, our argumentation would be that confidence prediction works well.
>
> ##### Question 2:  Virtual Node Clustering Implementation ...
>
> > The clustering process is done exclusively at inference time (we make this clear in the updated version of the manuscript). Since the number of virtual nodes used, is in almost all cases larger than the number of binding sites of the considered protein, it’s likely that several virtual nodes might get very close to the true binding site and to each other (with negligible numeric differences in the locations of the virtual nodes). Figure I1 (right) illustrates such a case, while Figure I1 (left) shows location predictions for another protein, which are spatially more distributed. In general, the case of using multiple virtual nodes to converge to nearby locations is more frequently observed for smaller proteins.
>
> ##### Question 3:  Practical Implications for Binding Site Identification ...
>
> > From a practical point of view, we expect that adjusting the number of virtual nodes towards a value, which is slightly larger than the number of usually observed binding sites, might be interesting to users. There could be strong binding interaction events with large confidence values, but maybe also weaker ones with lower confidence scores, which might however be still interesting for further investigation.
>
>
> ##### Question 4:  Overall a few sentences about the scaling in memory and computation according to protein size could be interesting ...
>
> > Since we work on neighborhood graphs and the neighborhood as well as the number of virtual nodes are usually quite limited (assumed to be a constant value), we expect that scaling properties behave linear in the protein size. We explicitly observed linear memory increase with a growing number of virtual nodes. The majority of the proteins in our dataset have not more than 8 annotated  binding sites, which are modeled as virtual nodes in our architecture and which justifies using a constant value of 8 virtual nodes. We provide insights on memory allocation for an increasing number of virtual nodes in Figure M1.
>
> ##### Question 5:  Finally, do the authors have any ideas ...
>
> > The reason why we specifically considered up to 5 layers, seems to have been caused by initial experimentation, where we followed the general paradigm "the deeper, the better" and obtained decent results, but we agree with the reviewer's thought that other hyperparameter choices could lead to competitive performance. The reason why five layers perform well might not be due to enhanced expressiveness, but rather could be because the learning behavior improves with additional layers. We add an additional ablation (see Figure L1) with different numbers of layers, which shows that decent results are achieved even with fewer layers.

---

> > ### Comment · Reviewer_Xvjk · 2024-11-25
> >
> > Dear Authors,
> >
> > Thanks for your reply! Overall I would say that your modifications and your comments have solved my questions. I will thus keep my rating as it is, which acknowledge that this is a good paper which should be accepted.

---

### Official Review · Reviewer_U9cY · 2024-11-01

**Soundness:** 2
**Presentation:** 3
**Contribution:** 2
**Rating:** 5
**Confidence:** 4

**Summary:**

**Summary:** The authors present VN-EGNNs for fast and accurate protein binding site prediction.

**Recommendation:** I am recommending a weak accept at this time.

**Rationale behind Recommendation:** If the authors were to explain their dataset splitting criteria more thoroughly and include additional experiments e.g., with reflection-sensitive scalar node features or e.g., with a virtual nodes variant of another type-1 equivariant graph neural network (e.g., GVPs [1]), I will consider raising my score.

**References:**

[1] Jing, Bowen, et al. "Learning from protein structure with geometric vector perceptrons." The Ninth International Conference on Learning Representations.

**Strengths:**

- Binding site identification presents a great opportunity to showcase the importance of virtual node learning.
- The computational efficiency of this approach is clear.
- The confidence predictions are a nice addition for this problem.
- VN-EGNN's t-SNE virtual node embedding plots show that the model has learned somewhat informative embeddings for proteins.
- The authors' discussion of achieving approximate invariance to the initial virtual nodes' positions through random augmentations is interesting.

**Weaknesses:**

- The authors' claim that VN-EGNNs are the first equivariant GNN equipped with virtual nodes does not seem to be accurate. In particular, recent works such as those of [1] seem to have explored this idea across a variety of molecular systems. I still think the authors' contributions for the problem of binding site identification are notable, however, it's worth noting that other works have already begun to explore extending equivariant GNNs with virtual nodes for improved expressivity. Please consider discussing works such as [1] to distinguish the authors' approach to developing an equivariant virtual nodes GNN.
- In Appendix G.1, the authors only describe their redundancy reduction (i.e., clustering) employed for the scPDB dataset. Please also describe how the authors have ensured their training and test splits for the PDBBind, COACH420, and HOLO4K datasets were not overlapping, to ensure fair benchmarking on the respective test splits.
- Based on their benchmarking results, the authors claim that residue-level information suffices for conformational binding site prediction. However, the authors' results still have room for improvement, so does this suggest that some methodological components are still missing? One such idea is that the ESM embeddings the authors employ are not sufficiently sensitive to chirality. Sensitizing the scalar node features to reflections (but not translations or rotations) may be worth exploring to see if this additional inductive bias of structural chirality improves VN-EGNN's results or not. Please see [2] for some ideas on how to make geometric GNNs sensitive to chirality, to consider if this improves VN-EGNN's performance for binding site identification.
- Concerning line 416, having to know the number of true binding sites in a protein to evaluate such each binding site predictor makes the benchmarking results in this work and in previous works less practically relevant. Please discuss the limitations and alternatives of this evaluation approach more carefully for readers.
- Regarding the fifth sentence of the authors' abstract, to increase the accessibility of this work, I'd recommend the authors consider rewriting this to read as something like, "However, the performance of GNNs at binding site identification is still limited potentially due to a lack of expressiveness capable of modeling higher-order geometric entities, such as binding pockets."
- If possible, releasing source code to accompany this model would be beneficial for the research community.

**References:**

[1] Zhang, Yuelin, et al. "Improving Equivariant Graph Neural Networks on Large Geometric Graphs via Virtual Nodes Learning." Forty-first International Conference on Machine Learning.

[2] Morehead, Alex, and Jianlin Cheng. "Geometry-complete perceptron networks for 3d molecular graphs." Bioinformatics 40.2 (2024): btae087.

**Questions:**

**Questions:**

- On line 290, should this read as "such that all eigenvectors..."?

**Feedback:**

- I think unsupervised geometric learning of other protein-related properties is a promising direction for future work.
- Typo on line 40: should be "a ligand" and not just "ligand".

---

> ### Author Response · Authors · 2024-11-23
>
> We thank the reviewer for providing a balanced review highlighting both the strengths and weaknesses, and for the possibility of potentially increasing the score. Below are our responses to the weaknesses and the question. We are grateful for making us aware of typos and even sharing thoughts about promising future directions.
>
> ##### Weakness 1: The authors' claim...
> > The work the reviewer mentioned has already referenced our research. At the time of their citation, our work was available in a preprint format.
>
> ##### Weakness 2: In Appendix G.1...
>
> > Here we adopted benchmarks from the community working on machine learning methods for binding site identification, especially from EquiPocket, which was recently published at this year’s ICML. This means that for comparability reasons, we used scPDB for training/hyperparameter selection and PDBBind, COACH420, and HOLO4K for testing. While we did not explicitly check for overlaps, it is worth mentioning that these datasets exhibit different characteristics. As shown in Figure J1 the only datasets comprising multiple chains are PDBBind2020 and HOLO4K. Further, the only datasets comprising multiple binding sites are COACH420 and HOLO4K. The training dataset does neither contain multiple binding sites nor multiple chains.
>
> ##### Weakness 3: Based on their...
>
> > The reviewer brings up an important topic: sensitivity to chirality. First of all it should be noted that the general scheme of VN-EGNN inherits the E(3) transformation equivariance/invariance properties of EGNN, which does especially mean predictions might get invariant to global reflections.
> For proteins, reflection invariance is not necessarily a desirable property since protein characteristics might change under reflection (which they usually are not considered to do under translation or rotation).
> To avoid reflection invariance in the context of protein graphs, it is sufficient to change node embeddings in case reflections occur, i.e., use other embeddings when the protein is reflected.
> In VN-EGNN, we encode amino acids by an ESM encoder. This encoder has been trained on tokens, which represent L amino acids. D amino acids have possibly been ignored in the ESM encoder, since they occur very rarely. We extend the ESM encoder by tokens representing D amino acids and initialize the embedding matrix randomly. Using an extended ESM encoder in that way, we encode L- and D-forms differently, and since amino acid residues change from an L-form to a D-form or vice versa under reflection, reflection invariance is broken.Thus reflection sensitivity is achieved.
> Additional comments:
> > - Such a random extension of an ESM encoder does not necessarily provide a very meaningful encoding of D amino acids, but allows to break reflection symmetry.
> > - On the other side there is lack of training data anyway for D amino acids, such that there is the question whether a more meaningful (than random) D amino acid encoding could be trained at all.
> > - Randomly extending the encoder for D amino acids should allow to break reflection symmetry for global reflections. In practice the occurrence of global reflections of proteins might however not at all be a relevant case. The aspect of chirality is more interesting with respect to local subparts of proteins. For this, global reflection symmetry of the architecture itself, is however not a problem.
>
> ##### Weakness 4: Concerning line 416...
> > We agree with the reviewer that requiring knowledge of the true number of binding sites for evaluation presents a general challenge in the field of binding site prediction. This limitation is indeed not specific to our work but affects all current approaches. However, this approach still allows a fair quantitative comparison between different methods in this research setting.
> The reviewer is right that there might be other more appropriate measures like IoU instead of computing a success rate with a predefined number of binding pockets.
>
> ##### Weakness 5: Regarding the fifth...
> > We thank the reviewer and adopt the sentence to the suggestion of the reviewer.
>
> ##### Weakness 6: If possible, releasing...
> > We provide an anonymized version of our source code at: https://anonymous.4open.science/r/vnegnn-code-3D77
>
> ##### Question:
> > Thanks for making us aware of that; we changed to "such that all eigenvalues of all other eigenvectors"

---

> ### Comment · Reviewer_U9cY · 2024-11-24
> **Response to author rebuttal**
>
> I would like to thank the authors for their rebuttal to my initial review. In general, my opinion of this paper is unchanged: I think the practical contributions of this work are clearly demonstrated through the authors' experiments, though from a theoretical (and data curation) perspective, there is certainly room for improvement for this problem domain. Even though previous works have used datasets that are not necessarily filtered to strictly reduce overlap between training and test splits, I think the authors should consider revisiting this concern in follow-up work if possible. As such, I will hold my score at a weak accept (6) for now.

---

### Official Review · Reviewer_nVpp · 2024-11-02

**Soundness:** 2
**Presentation:** 1
**Contribution:** 1
**Rating:** 3
**Confidence:** 5

**Summary:**

This paper proposes an extended version of EGNN called VN-EGNN by introducing virtual nodes and applying extended message passing scheme. It focuses on protein binding site identification problems and claims achieving SOTA at locating binding site centers on COACH420, HOLO4K and PDBbind2020 datasets.

**Strengths:**

The problem studied in this paper is very important in the field of protein. The proposed VN-EGNN is very simple, and I believe that this type of method has very good scalability and can be further extended to other equivariant networks.

**Weaknesses:**

>**W.1 The entry point of the article is rather vague.**

The motivation of this paper is to develop a specialized design, but the methods adopted are more inclined towards general approaches. In the abstract, the authors state, 'The virtual nodes in these graphs are dedicated entities to learn representations of binding sites, which leads to improved predictive performance.' However, in the actual design, it resembles a general method and does not incorporate prior knowledge specific to the mechanism of virtual nodes (only using the loss function).
It is unclear whether the authors intend to emphasize the application value of this work (specialization) or its theoretical value (generality). If it is the former, more prior knowledge should be introduced for specialized design, and the biochemical significance should be analyzed; otherwise, this virtual node approach should be extended to other models, such as SchNet [a], PAINN [b], TFN [c], Allegro [d], and MACE [e].

[a] Schnet – a deep learning architecture for molecules and materials.
[b] Equivariant message passing for the prediction of tensorial properties and molecular spectra.
[c] Tensor field networks: Rotation-and translation-equivariant neural networks for 3d point clouds.
[d] Learning Local Equivariant Representations for Large-Scale Atomistic Dynamics.
[e] Mace: Higher order equivariant message passing neural networks for fast and accurate force fields.

> **W.2 The methodology of the article is inconsistent with that used in the actual experiment.**

The main purpose of this paper is to vacillate between designing a strict equivariant network and allowing the network to be approximately equivariant, which is very fragmented and confusing. In the section "Equivariant initialization of virtual nodes in VN-EGNN.", the virtual nodes are initialized at the center of the whole graph, which is a strictly equivariant operation. In the section "Data augmentation and approximate equivariance", the Fibonacci grid, an unequivariant operation, is used. This makes the paper inconsistent and many chapters lose their meaning, including the proof of equivariance (Appendix E) and the analysis of expressiveness (Appendix K), see comment W.5.

> **W.3 The contribution of the article is not enough.**

Adding global information such as virtual nodes (or even meshes) to the graph is a very obvious idea, and many previous works have studied it, including both non-equivariant and equivariant ones. The most classic non-equivariant work (such as MPNN [f]), and equivariant work includes using priors (such as MEAN [g], AbDiffuser [h]) and not using priors (such as FastEGNN [i], Neural P^3M [j]). The article claims that "To the best of our knowledge VN-EGNN is the first E(3)-equivariant GNN architecture using virtual nodes.", but ignores these pioneering works. Importantly, the core contribution of all the above articles is not virtual nodes, but treat the virtual nodes as only a useful engineering trick. The simple contribution of introducing virtual nodes is not enough to make it accepted (not to mention that such an introduction seems to have some hidden dangers). The author can consider the suggestions in the two directions in W1 to modify the article.
[f] Neural message passing for quantum chemistry.
[g] Conditional Antibody Design as 3D Equivariant Graph Translation.
[h] AbDiffuser: full-atom generation of in-vitro functioning antibodies.
[i] Improving Equivariant Graph Neural Networks on Large Geometric Graphs via Virtual Nodes Learning
[j] Neural P^3M: A Long-Range Interaction Modeling Enhancer for Geometric GNNs

> **W.4 The experiment is not convincing enough.**

(i) The article should compare relevant articles (such as those mentioned in W.3) and more lastest models (e.g. SphereNet [k], ClofNet [l], LEFTNet [m], ViSNet [n], Geoformer [o], SO3krates [p]) as baselines.
(ii) In addition, according to Appendix F, VN-EGNN performs random rotations during training, which is unfair to equivariant networks. This approach does not show that such invariance/equivariance is brought by VN-EGNN itself, but may be due to data augmentation. Instead of using this approach, it is better to discard the restrictions of the architecture and learn the potential equivariance completely through data-driven learning, like AlphaFold3 [q].
(iii) The dataset selected in the article contains substances of various conformations, which is equivalent to data augmentation. It should be ensured that the input conformations of the training set are similar to each other (such as protein dynamics in FastEGNN [i]), and then only the validation set and test set are randomly rotated to verify whether the model is strictly or approximately equivariant.
[k] Spherical Message Passing for 3D Molecular Graphs.
[l] SE(3) Equivariant Graph Neural Networks with Complete Local Frames.
[m] A new perspective on building efficient and expressive 3d equivariant graph neural networks.
[n] Enhancing geometric representations for molecules with equivariant vector-scalar interactive message passing.
[o] Geometric transformer with interatomic positional encoding.
[p] A Euclidean transformer for fast and stable machine learned force fields.
[q] Accurate structure prediction of biomolecular interactions with AlphaFold 3.

> **W.5 Problems caused by non-strict equivariance.**

(i) The equivariant model requires equivariance at each layer. Using the Fibonacci grid method to initialize virtual nodes will make the equivariance of the entire model no longer meaningful.
(ii) VN-EGNN uses non-equivariant initialization, which seems to enhance the expressive power of equivariant neural networks, but actually reduces the robustness of the model. Let's assume that the first virtual node initialized by the Fibonacci grid is $(x,y,z)$ and is not at the origin. The two graphs are $\\{(\pm 1, 0,0)\\}$ and $\\{(0, \pm 1, 0)\\}$. Obviously, the two graphs are geometrically isomorphic, but the virtual nodes introduced by VN-EGNN cannot guarantee the same output.

> **W.6 Relationship between Fibonacci grid and number of virtual nodes.**

Fibonacci grid is a commonly used technique for generating approximate equivariance, and plays an important role in eSCN [r] and EquiformerV2 [s]. However, Fig. 9 in eSCN also points out that to make the equivariance error very low, $18\times 18=324$ samplings may be required, corresponding to the virtual nodes of VN-EGNN. In fact, VN-EGNN only uses 4 or 8 virtual nodes, which makes people very worried about whether it will bring a large equivariance error. And if VN-EGNN really uses 324 virtual nodes, will the overhead become unacceptable and lose good scalability?
[r] Reducing SO(3) Convolutions to SO(2) for Efficient Equivariant GNNs.
[s] EquiformerV2: Improved Equivariant Transformer for Scaling to Higher-Degree Representations.

**Questions:**

See Weakness.

---

> ### Author Response · Authors · 2024-11-23
>
> We thank the reviewer for the encouraging words on scalability and hope that our responses to weaknesses might help to reconsider their initial assessment and view our paper more favorably for acceptance.
>
> ##### Weakness 1:
>
> > The reviewer is right that the suggested GNN architecture might also be suited for other applications than just Protein Binding Site Identification. An important aspect in creating a machine learning model with high prediction accuracy is however also the design of the **neural network architecture** itself as well as associated **new learning schemes (extension of loss function, extension of learning algorithm)**, . We consider exactly this to be the **exploitation of prior knowledge** as questioned by the reviewer. In the manuscript we tried to argue why exactly this network architecture might make a lot of sense for protein binding site identification and we would like to point out that more generic network architectures such as GAT, **SchNet**, or, EGNNs showed worse performance.
> >
> > It is important to acknowledge that there is often a trade-off between generic learning approaches and those that are more specialized and application-specific. The effectiveness of our suggested network architecture may be attributed to the limited amount of available training data in the field of protein binding site identification, which might make it necessary to use a more specialized network architecture to obtain state-of-the-art results.  Whether the usage of virtual nodes and an associated loss function as we suggested, e.g., might be competitive for application fields with hundreds of thousands of data points is questionable.
> >
> > For the mentioned reasons, we would like to remark that we (1) consider our paper to be an application paper (i.e., we submitted it to ICLR to the category “applications to physical sciences (physics, chemistry, biology, etc.)”), and, (2) that we discuss the limitation to the specific application domain in the Limitations-section of the paper. **We add to the limitations section, that the reason why our network architecture works especially well for binding site identification might be the relatively low number of training data points.**
>
>
> ##### Weakness 2:
>
> > We try to improve the text. What we actually wanted to bring over is:
> > - From an application point of view, having strict equivariance is a desirable property for a binding site identification algorithm in principle. In its core our suggested architecture is able to fulfill this property.
> > - However from a practical point of view it seems advantageous not to restrict the learning architecture too much. Therefore, we came up with a more approximate version combined with data augmentation (random rotations during training) of our architecture, which we are using in practice. We found this approach led to a more diverse spatial distribution across the protein surface while maintaining model performance.
> > - In a wider context, we would like to remark that AlphaFold 3 dropped model architecture restrictions with respect to equivariance for protein structure predictions. Compared to AlphaFold 2, which used a more strict form of built-in equivariance, structure prediction performance could be further increased. From a general point of view however, having SE(3)-equivariant predictions is still a desirable property for structure prediction.
>
>
>
> ##### Weakness 3:
>
> > We are happy to cite MEAN as the potentially first virtual node method in the context of EGNNs. **We thank the reviewer for making us aware of the publication and will remove that "To the best of our knowledge VN-EGNN is the first E(3)-equivariant GNN architecture using virtual nodes."** The aim of our work is similar to that of MEAN in using an adapted network architecture for a specific application in mind. We thereby tried to avoid some of the disadvantages (as, e.g., discussed in Joshi et al, 2023) of vanilla EGNNs and used a neural representation layer update scheme which might not have existed before. We were not sure, whether we could correctly identify the part, where the reviewer found an analogy of our work to AbDiffuser, but are open to cite it, in case the reviewer points us more directly to this analogy. One further work, which the reviewer mentioned, has already referenced our research. At the time of their citation, our work was available in a preprint format. Another suggested further work, mentioned by the reviewer, missed to reference this preprint.

---

> > ### Author Response · Authors · 2024-11-23
> >
> > ##### Weakness 4:
> >
> > > We agree with the reviewers that the suggested methods might serve as interesting base architectures for Binding Site Identification, especially due to their potentially increased expressivity. It should however be considered, that the choice of a specific GNN or the choice of a specific message passing architecture is only one important aspect in designing a Binding Site Identification method. Other important aspects, e.g.,  concern the representation of amino acid properties or whether the protein is solely represented by residues or by all its atoms. In our submitted research, we found that EGNN seems to serve as a decent message passing architecture, which we could successfully extend towards a binding site identification method, which achieves **new state-of-the-art results**. We therefore consider the adaption of other and potentially **more expressive message passing schemes as a promising direction towards further improvements** for Binding Site Identification, especially in the light of faster GPUs with more memories in future. Adapting the mentioned architectures and providing results for them is however beyond the scope of the current submission. We agree with the reviewer, that a main reason for the success of VN-EGNN might not only be the invariance/equivariance brought in by the architecture, but also the applied data augmentation scheme. **We are sorry** in case we might have given the impression that solely invariance/equivariance brought in by the architecture is responsible for the success and change the manuscript accordingly (see our changes in the Contributions part). We **appeal to the reviewer** to reconsider, that our **primary research goal** was the development of a new state-of-the-art Binding Site Identification method and not to suggest a generic GNN architecture with exhibiting advantageous properties.
> >
> > ##### Weakness 5:
> >
> > > The reviewer is **completely right** and shows with the given example problems, which might occur with initialization by the Fibonacci grid. There however seem to be reasons, why our method might still work in practice for the given task and gives similar outputs:
> > > - First, we use 8 virtual nodes. Due to using more than one virtual node, which are well distributed by the Fibonacci grid, large differences between different initializations might cancel out.
> > > - Our training strategy might help, that our model learns to overcome slight differences in initializations during the iterative VN-EGNN update procedure (consider it has more than one layer).
> > > - **Most importantly**: For the protein task we show variances resulting from an ablation experiment with different initializations at the inference stage in Table F1. We could indeed show that differences in initialization have **only minor effects on the final predictions**.
> > Our observations seem to be in accordance with findings on equivariant representation learning by others. The authors of https://arxiv.org/abs/2410.17878 relax equivariance and nevertheless observe learning “approximate symmetries by minimizing an additional simple equivariance loss”.\
> > \
> > Why is equivariance inherited by the scheme of EGNN then still useful for us? https://arxiv.org/abs/2410.23179 mentions, that “equivariance improves data efficiency, but training non-equivariant models with data augmentation can close this gap given sufficient epochs”.\
> > \
> > Although, with the initialization by the Fibonacci grid, we are not fully equivariant any more, we keep as many equivariance properties throughout our approach as possible, while not restricting our architecture too much.
> >
> >
> > ##### Weakness 6:
> >
> > > The reviewer is right, that scalability is an issue, which is why the number of virtual nodes is relatively small in VN-EGNN. We think the difference to eSCN is that the Fibonacci grid in eSCN is used for **a different purpose**, i.e., to **approximate integrals** in order to compute energies and forces, which might require a larger number of points to get numerically accurate results. We used up to 8 virtual nodes since this is in almost all cases already **larger than the usual number of observed binding sites** seen in wet-lab experiments. The task in our application is to find these locations at a protein. Small numeric equivariance errors nevertheless seem allowing us to find these locations with high probability.

---

> > > ### Comment · Reviewer_nVpp · 2024-11-24
> > > **Discussion (1/2)**
> > >
> > > The author's reply does not resolve my doubts, so I keep my score. The author seems to deliberately confuse the topic of architecture-based equivariance with data-driven equivariance, and there are no additional experiments in the reply. In addition, the article cited by the author seems to prove that my concerns are correct.
> > >
> > > >D1. Comparison with FastEGNN (also reply to reviewer U9cY)
> > >
> > > Reviewer U9cY and I both mentioned FastEGNN, which is a work that strictly satisfies equivariance, and its motivation (for large-scale geometric graph acceleration) and message passing mechanism are completely different and much more sophisticated than this paper (including theoretical analysis and experimental verification). In addition, the author mentioned that FastEGNN cited the preprint of this paper. I would like to point out that FastEGNN's citations to VN-EGNN are concentrated in Table. 8 in the appendix, and they are **negative citations**. Its experimental results verify that the architecture of VN-EGNN is not equivariant, and can only rely on data enhancement to achieve similar results. The authors should explain the experimental results.
> > >
> > > In addition, since the author mentioned the preprint of FastEGNN and this paper, I checked and found that in the preprint, only the use of Fibonacci grid for initialization was mentioned, rather than the current CoM initialization (used in FastEGNN). From this point of view, the authors should cite FastEGNN and further explain how VN-EGNN is different from it.
> > >
> > > >D2. Unfair comparison between VN-EGNN and other baseline models
> > >
> > > In the training of VN-EGNN, you used data augmentation (random rotation during training). However, the experimental results of almost all other baseline models in Table 1 come from EquiPocket, which does not use any data augmentation in its test. Therefore, the comparison between VN-EGNN and other baseline models in the article is unfair. If you want to use the results of EquiPocket, you need to not rotate during training and rotate during testing; if you use data augmentation during training, you need to adopt the same method when training other models (including but not limited to Fpocket, DeepSite, Kalasanty, DeepSurf, GAT, GCN, GAT+GCN, GCN2). You cannot directly use the results of EquiPocket.
> > >
> > > >D3. About architectural contribution, application contribution and GWL test.
> > >
> > > The authors should not call for attention to the application contributions of this paper only when the reviewers do not see the architectural problem, and promote the equivariance of this paper at other times. Not fully equivariant is not the same as being somewhat equivariant. Data-driven equivariance is unreliable and is not enough to be an innovative point for the paper to be accepted. In fact, as early as the development of computer vision, data augmentation through random rotations can also be interpreted as data-driven equivariance.
> > >
> > > It must be admitted that the binding problem studied in this paper is indeed very valuable, and the idea of using virtual nodes is also natural. The article emphasizes the significance of architectural equivariance that does not actually exist, which is unreasonable, and I think the description of the article must be revised. If the authors can modify the claim of the contribution to equivariance and focus the article on application, I will **increase my score**. From the perspective that the idea of using virtual nodes to correspond to binding sites is indeed interesting, and FastEGNN also admits in its Fig. 1 that it cannot do this (i.e., it can only partially reflect the motion mode, not having special biochemical significance), I think the author can explain the difference between the two articles from this perspective.
> > >
> > > In addition, the GWL results in the preprint by the author actually brought bias to the evaluation of other articles. For example, ETNN [a], I checked its code and found that it actually adopted the CoM initialization method, but cited VN-EGNN. Although ETNN's ability to distinguish GWL test will not be at the cost of possible misjudgment, if other articles continue to cite VN-EGNN with incorrect initialization, this will cause a large number of models that cannot distinguish to appear to have the ability to distinguish. The author should explain the experiment as soon as possible to avoid further expansion of the problem.
> > >
> > > [a] E(n) Equivariant Topological Neural Networks
> > >
> > > >D4. Supplement experiments using CoM initialization
> > >
> > > In order to make the logic of the article smooth, it is necessary to supplement the experiment of CoM initialization, and it is necessary to update the illustrations of the article.

---

> ### Comment · Reviewer_nVpp · 2024-11-24
> **Discussion (2/2)**
>
> >D5. Utilization efficiency of virtual nodes
>
> From Figs. 2 and I1, we can see that there may be multiple virtual nodes converging to the same binding site. Can we improve the utilization efficiency of virtual nodes by introducing MMD Loss or other optimal transmission Loss like FastEGNN?
>
> >D6. Difference between E(3)-equivariance and SE(3)-equivariance
>
> In fact, if the authors look up the entry on chirality in Wikipedia [b], they will find that chirality does not affect physical properties. The difference in biochemical properties is due to the fact that the human body does not undergo the same Euclidean transformation, that is, the entire system is not geometrically isomorphic to the original system. The task of this article does not seem to have such a quantity, so I think the chirality explanation here is far-fetched.
>
> [b] https://en.wikipedia.org/wiki/Chirality_(chemistry)

---

> > ### Author Response · Authors · 2024-11-24
> >
> > **Architectural vs learned equivariance:**
> > > VN-EGNN supports both architecture-based equivariance and data-driven equivariance via different initialization strategies -- note that there is no "correct" choice here, but these are both valid and frequently used machine-learning approaches. For binding site identification, data-driven equivariance with random rotations yielded better performance, as stated in the manuscript.
> >
> > **Fair Comparisons:**
> > > Each method can preprocess or modify input data as preferred (e.g. also DeepSurf performs rotations). Our comparisons fairly reflect the results achievable under this flexibility.
> >
> > **D5**
> > > This is indeed a good point that the reviewer raises. We will mention this as future directions for this work.
> >
> > **D6**
> > > The Wikipedia article oversimplifies. Stereoisomers, especially L- and D-amino acids, often have distinct biochemical properties, crucial in biological contexts like protein-ligand interactions.

---

> > > ### Comment · Reviewer_nVpp · 2024-11-25
> > >
> > > I have no intention of dwelling on the authors' word games. They did not directly answer any of my questions. I ask the authors to directly answer the following points:
> > >
> > > 1. Does this paper fail to achieve strict equivariance in architecture? If it does not, please reply to the next question; if it does, please prove it.
> > >
> > > 2. Can this paper guarantee that the E(3)/SE(3)-equivariance driven by data is reliable? If not, please reply to the next question; if so, please give a theoretical upper bound or experimental curve of equivariance loss and explain the experimental results of VN-EGNN's loss explosion under random rotation in Table 8 of FastEGNN.
> > >
> > > 3. Which of the equivariance in this paper is caused by the architecture? Which is caused by data-driven? In other words, which part of the equivariance loss is caused by the architecture and which part is caused by data-driven? Please give a theoretical analysis.
> > >
> > > If the authors can clearly answer any of these three questions, I will immediately raise my score to **"clearly accepted"**. If the authors cannot answer any of them, then I cannot understand what contribution the authors have made in terms of equivariance.
> > >
> > > These three questions are not deliberately difficult. All strictly equivariant models can answer the first point, such as EGNN, PAINN, FastEGNN, TFN, SEGNN, MACE; all approximately equivariant models can answer the second point, such as eSCN, EquiformerV2; I have not seen any work that can answer the third point. If the authors can answer it, it will be a milestone contribution, and I will further improve my score to **"strongly accepted"**.
> > >
> > > The following are my suggestions if the authors cannot answer any of them. This paper should be distinguished from equivariant models in writing, including but not limited to:
> > > - Compare with the operation of introducing virtual nodes in the strict equivariance of FastEGNN (including theoretical analysis and experimental analysis).
> > > - Provide data enhancement for all models in the experiment, or abandon the data enhancement of VN-EGNN to ensure the fairness and credibility of the experimental results.
> > > - Remove the experimental results of GWL-test at the cost of possible misjudgment to avoid further erroneous influence.
> > >
> > > If the author can do this, I still think it can be accepted due to its excellent application value.

---

> > > > ### Author Response · Authors · 2024-11-27
> > > >
> > > > We greatly appreciate the time and effort dedicated to reviewing our work and want to answer the open questions.
> > > >
> > > > Based on the reviewer’s feedback, we ran one additional experiment (A) and we also added a learning curve (B) on training a VN-EGNN binding site model with Center-of-Mass (CoM) initialization. We now cite FastEGNN to acknowledge that CoM was initially proposed in their work.
> > > >
> > > > - (A) We added a k-chains experiment with additional CoM-initialization. Also with the CoM-initialization, the problems can be solved.
> > > >
> > > > - (B) We added the learning curve of an initial training run for creating a binding site identification model with CoM initialization, which was the reason to relax the strict equivariance. In the plot we also show how learning curves for training models with Fibonacci-grid initialization together with data augmentation behave.
> > > >
> > > > Questions:
> > > >
> > > > Q1: Strict equivariance.
> > > > > Yes, if VN-EGNN is run with Center-of-Mass (we now cite ref [1] and [2] here) initialization, VN-EGNN **maintains** strict equivariance (proof is trivial since the center of mass is equivariant to rotations and translations). No, if VN-EGNN is run with the Fibonacci-grid initialization and data augmentation, it does not employ strict equivariance. For the binding site identification task, we use VN-EGNN with Fibonacci-grid initialization and data augmentation, because it performs better (see Figure H2). We think this initialization offers an effective approach for the binding site identification problem, since the initial virtual node embeddings are directly derived from the protein. (see Line 456).
> > > >
> > > > Q2: Reliability.
> > > > > With the data-driven approximate equivariance, we show in Section F.1 and Table F1 that the performance of VN-EGNN remains stable under rotations of the input. Also in an additional experiment with k-chains, we show that with two different initialization schemes, the results remain similar (Table K1).
> > > >
> > > > Q3: Relations between architecture and equivariance.
> > > > > The VN-EGNN architecture itself guarantees equivariance after the virtual nodes are initialized. Depending on the initialization and data augmentation strategy, VN-EGNN can either have exact equivariance (with CoM [1,2] initialization) or approximate equivariance with Fibonacci-grid initialization and data augmentation. Thus, we clearly know from which components of the methods the equivariance properties arise.
> > > >
> > > > We hope that our response, together with the additional experimental results, highlights the application value of our work for the reviewer.
> > > >
> > > > Updated sections:
> > > >
> > > > - Line 269 reference to FastEGNN
> > > > - Line 1380 additional Figure H2
> > > > - Line 1642 extended k-chains experiment with CoM
> > > >
> > > > References:
> > > >
> > > > [1] Zhang, Y., Cen, J., Han, J., Zhang, Z., Zhou, J., & Huang, W. Improving Equivariant Graph Neural Networks on Large Geometric Graphs via Virtual Nodes Learning. In Forty-first International Conference on Machine Learning.
> > > >
> > > > [2] Kaba, S. O., Mondal, A. K., Zhang, Y., Bengio, Y., & Ravanbakhsh, S. (2023, July). Equivariance with learned canonicalization functions. In International Conference on Machine Learning (pp. 15546-15566). PMLR.

---

### Official Review · Reviewer_YaLZ · 2024-11-02

**Soundness:** 2
**Presentation:** 3
**Contribution:** 3
**Rating:** 5
**Confidence:** 4

**Summary:**

This study improves E(n)-equivariant graph neural networks (EGNNs) framework to predict protein-ligand biding site through two innovations: 1) proposing virtual nodes; 2) applying an extended message passing approach. The performance of the proposed approach has been benchmarked with other baselines on three data sets.

**Strengths:**

Overall, I think the paper is well-written.
1. Although the topic is classical and the virtual node idea is also not novel, which has been explored in many other fields,  the improvement on the prediction performance using the two proposed ideas (virtual nodes and the improved message passing) are obvious.
2. The proofs of the properties of the proposed approach are interesting and solid.
3. The research is comprehensive and clear, and the citations of the paper are detailed.

**Weaknesses:**

1. In the result tables, since you have calculated standard deviations, you can also calculate the p-values to measure whether the performance of your model is significantly different from the baselines in the different datasets.
2. Need some details about the prediction of the data set (HOLO4k) with domain shift. Was the same approach applied? why or why not can the proposed method handle the domain shift issue?


*************
During the discussion among reviewers and area chairs, it became apparent that the performance comparison between the proposed method and the baseline was unfair. The proposed method relies on augmented data, while the baseline does not. As a result, the score was adjusted to 5.

**Questions:**

1. I am wondering if you can generate a visualization plot to show if the predicted binding site(s) have high weights in your trained model(s), which can make the model more interpretable.

**Details Of Ethics Concerns:**

Not applicable.

---

> ### Author Response · Authors · 2024-11-23
>
> We thank the reviewer for acknowledging the strengths of our research and hope that our response on the brought up weaknesses and questions can address the raised concerns and questions.
>
> ##### Weakness 1:
> > Yes, when we apply a one-sided Wilcoxon test, with the null hypothesis that VN-EGNN is worse with respect to the DCC metric than the scond-best method (P2Rank), we obtain p-values < 0.05 on the datasets PDBBind and COACH420. For HOLO4K the p-value is about 0.06. When we apply the same test for the DCA metric instead of the DCC metric, we do not observe significantly better prediction performances. The test results are in accordance to Table 1, for which we slightly changed the definition of the bold markings to be in accordance with the suggested tests.
>
>
> ##### Weakness 2:
> > Yes, we applied the same approach. The domain shift here is that HOLO4k predominantly contains multi-chain protein complexes with multiple binding sites, which is in contrast to the scPDB training dataset with single-chain proteins. To be able to provide binding site predictions for multi-chain protein complexes with k (experimentally resolved) chains in a meaningful way, we first split the protein up into multiple chains. We then apply our model, which was primarily trained on single chain proteins, individually to each single chain, i.e., we apply our model k times to get 8 predictions (where 8 is the number of virtual nodes, which is a hyperparameter of our model) for each protein chain together with the self-confidence scores for the individual virtual nodes. Finally, we merge the predictions and sort the 8*k predictions by their scores according to our confidence model.
> >
> > To evaluate, we consider the k last (i.e., the k highest-ranked) predictions to be the identified binding sites by VN-EGNN. This is in accordance to the performance evaluation of Equipocket and maintains comparability of the results.
>
> ##### Question:
> > A figure (former Figure I1) in the appendix should have shown the correlation between predicted virtual node scores and proximity to the true binding pocket, with nodes nearest to the actual pocket receiving higher scores. We now moved this figure into the main part of our manuscript to make an interpretation how well our method might perform more accessible to readers.

---

> > ### Comment · Reviewer_YaLZ · 2024-11-25
> >
> > Thank you for the clarification. Most of my comments have been addressed, and I am now happy to increase my score to 8.

---

### Official Review · Reviewer_MytV · 2024-11-04

**Soundness:** 2
**Presentation:** 2
**Contribution:** 2
**Rating:** 5
**Confidence:** 3

**Summary:**

This paper focuses on enhancing binding site identification in proteins using extended E(n)-equivariant graph neural networks (EGNNs) with the introduction of virtual nodes. Traditional EGNNs have struggled with this task due to the absence of dedicated nodes for learning binding site representations and issues related to oversquashing in message passing. The proposed VN-EGNN method aims to address these challenges, demonstrating significant improvements in predictive performance across several benchmark datasets (COACH420, HOLO4K, and PDBbind2020). The paper provided a comprehensive overview of the problem, related work, and the proposed methodology.

**Strengths:**

1. The introduction of virtual nodes to enhance the learning of binding site representations addresses key limitations in traditional EGNNs, offering a novel method for protein binding site identification.
2. VN-EGNN demonstrates the state-of-the-art performance in binding site identification across multiple benchmark datasets.

**Weaknesses:**

1. In the ablation experiments, the ablation of virtual nodes should ensure the presence of heterogeneous message passing and the pre-trained protein embedding module, rather than the traditional EGNN.
2. In line 135, the coordinates of the virtual nodes are initialized randomly, and they ultimately converge to the coordinates of the ligands. So why are two initialization strategies for the virtual nodes mentioned in section 2.3? Could you provide more discussion on the initialization of the positions of the virtual nodes?
3. In the design of the loss function, Dice loss was used for binding site identification as a node-level prediction task. How does VN-EGNN perform on this task?

**Questions:**

see above

---

> ### Author Response · Authors · 2024-11-23
>
> We thank the reviewer for their effort and think that we can address the reviewer's concerns sufficiently to recommend acceptance.
>
> ##### Weakness 1:
> > The reviewer is right that this is the more interesting ablation. We provide exactly this ablation as “VN-EGNN (homog.)” as an ablation to “VN-EGNN (full)”. We rename “VN-EGNN (VN only)” to “EGNN+VN” to make clear that this ablation is just plain EGNN with virtual nodes. We also try to make the related text more clear and thank the reviewer for pointing that up.
>
> ##### Weakness 2:
> > We try to improve the text. What we actually wanted to bring over is:
> > - From an application point of view, having strict equivariance is a desirable property for a binding site identification algorithm in principle. In its core our suggested architecture is able to fulfill this property.
> > - However from a practical point of view it seems advantageous not to restrict the learning architecture too much. Therefore, we came up with a more approximate version combined with data augmentation (random rotations during training) of our architecture, which we are using in practice. We found this approach led to a more diverse spatial distribution across the protein surface while maintaining model performance.
>
>
>
> ##### Weakness 3:
> > We thank the reviewer for the suggestion and provide it in Table H1.
> >
> > |             | Dice loss     | IOU           |
> > | :---------- | ------------: | ------------: |
> > | COACH420    | 0.397 (0.015) | 0.437 (0.005) |
> > | HOLO4K      | 0.584 (0.031) | 0.263 (0.025) |
> > | PDBBind2020 | 0.357 (0.010) | 0.477 (0.003) |

---

> > ### Comment · Reviewer_MytV · 2024-11-26
> >
> > I appreciate the authors' detailed rebuttal, and I have decided to maintain my score, as I remain concerned about the authors' contributions to addressing the problem.

---

### Comment · Area_Chair_FPXZ · 2024-11-25

Dear Reviewers,

The authors have uploaded their rebuttal. Please take this opportunity to discuss any concerns you may have with the authors.

AC

---

### Meta-Review · Area_Chair_FPXZ · 2024-12-20

**Metareview:**

This paper introduces VN-EGNN, an extension of E(n)-equivariant graph neural networks (EGNNs), which incorporates virtual nodes to improve protein binding site identification. The method attempts to address issues in traditional EGNNs, such as binding site representation and message passing inefficiencies, and demonstrates good performance on several benchmark datasets.

While the proposed VN-EGNN model shows promising results, the reviewers have identified several weaknesses that need to be addressed:

1. The use of Fibonacci grid initialization undermines the model’s equivariance, reducing it to a version of FastEGNN, which lacks innovation and is not strictly equivariant
2. The use of random rotations during training introduces data augmentation that unfairly distorts the comparison with baselines, which did not use similar augmentation, making the experimental results incomparable.
3. The GWL-test results, which demonstrate VN-EGNN’s ability to distinguish structures, do not indicate stronger expressivity, as successful differentiation comes at the cost of misclassification, and CoM initialization reduces the model to FastEGNN.

Based on these weaknesses, we recommend rejecting this paper. We hope this feedback helps the authors improve their paper.

**Additional Comments On Reviewer Discussion:**

The authors’ rebuttal emphasizes their efforts to improve clarity and provide detailed responses to reviewers’ concerns, including structural improvements, language refinements, and additional analyses. They also address related work, make their code publicly available, and highlight key changes in the manuscript.

However, during the discussion phase, several reviewers raised concerns about both the experimental setup and the theoretical contributions of the paper, leading them to lower their scores. Based on their feedback, I recommend rejecting the paper.

---

### Decision · Program_Chairs · 2025-01-22

Reject